# A Light-robust Reconstruction Method for Spike Camera

## Abstract

Spike camera with high temporal resolution can fire continuous binary spike streams to record per-pixel light intensity. By using reconstruction methods, the scene details in high-speed scenes can be restored from spike streams. However, existing methods struggle to perform well in low-light environments due to insufficient information in spike streams. To this end, we propose a recurrent-based reconstruction framework to better handle such extreme conditions. In more detail, a **l**ight-**r**obust **rep**resentation (LR-Rep) is designed to aggregate temporal information in spike streams. Moreover, a fusion module is used to extract temporal features. Besides, we synthesize a reconstruction benchmark for high-speed low-light scenes where light sources are carefully designed to be consistent with reality. The experiment shows the superiority of our method. Importantly, our method also generalizes well to real spike streams. All codes and constructed datasets will be released after publication.

## 1 Introduction

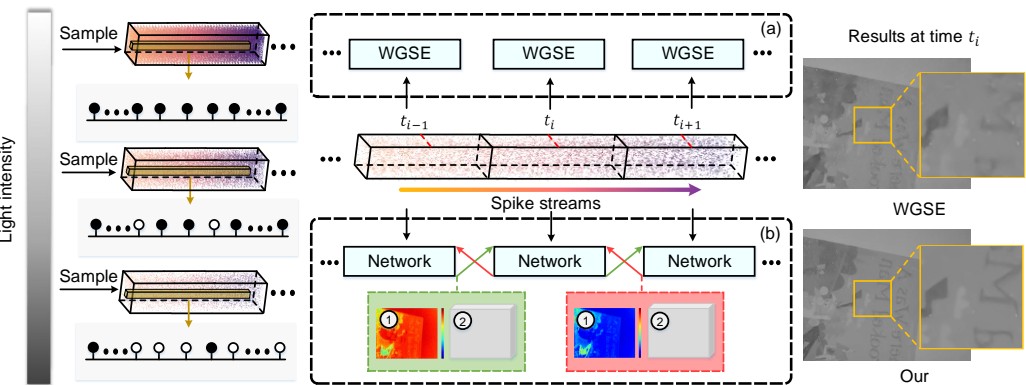

Figure 1: Overview of reconstruction for high-speed spike streams. Left: with decreasing light intensity, more sparse spike streams are difficult to extract features. A black circle is a spike. Middle: (a) The state-of-the-art method, WGSE (Zhang et al., 2023). The arrow with a gradient color is the timeline. (b) Our reconstruction method. Green (red) lines denote the forward (backward) data flow. ① (②) is the release time of spikes (temporal features). ① (②) in forward and backward data flow is independent. Right: reconstructed results from WGSE and our method.

As a neuromorphic sensor with high temporal resolution (40000 Hz), spike camera (Zhu et al., 2019; Huang et al., 2022) has shown enormous potential for high-speed visual tasks, such as reconstruction (Zhao et al., 2020; Zhu et al., 2022; Zheng et al., 2021; Zhao et al., 2021; Zhu et al., 2021; chen et al., 2022), optical flow estimation (Hu et al., 2022; Zhao et al., 2022b; Zhai et al., 2023), and depth estimation (Zhang et al., 2022; Liu et al., 2022; Li et al., 2022). Different from event cameras (Lichtsteiner et al., 2008; Delbrück et al., 2010; Brandli et al., 2014), it can record per-pixel light

intensity by accumulating photons and firing continuous binary spike streams. Correspondingly, high-speed dynamic scenes can be reconstructed from spike streams. Recently, many deep learning methods have advanced this field and shown great success in reconstructing more detailed scenes. However, existing methods struggle to perform well in low-light environments due to insufficient information in spike streams.

A dilemma arises for visual sensors, that is, the quality of sampled data can greatly decrease in a low-light environment (Guo et al., 2020; Li et al., 2021c;b; Zhao et al., 2022b; Graca et al., 2023). Low-quality data creates many difficulties for all kinds of vision tasks. Similarly, the reconstruction for the spike camera also suffers from this problem. To improve the performance of reconstruction in low-light high-speed scenes, two non-trivial matters should be carefully considered. First, Constructing a low-light high-speed scene benchmark for spike camera is crucial to evaluating different methods. However, due to the frame rate limitations of traditional cameras, it is difficult to capture images clearly in real high-speed scenes as supervised signals. Instead of it, a reasonable way is to synthesize datasets for spike camera (Zhao et al., 2021; Zhu et al., 2021; Hu et al., 2022; Zhang et al., 2022). To ensure the reliability of the reconstruction benchmark, synthetic low-light high-speed scenes should be as consistent as possible with the real world, *e.g.,* light source. Second, as shown in Fig. 1, with the decrease of illuminance in the environment, the total number of spikes in spike streams decreases greatly which means the valid information in spike streams can greatly decrease. Fig. 1(a) shows that previous methods often fail under low-light conditions since they have no choice but to rely on inadequate information.

In this work, we aim to address all two issues above-mentioned. In more detail, a reconstruction benchmark for high-speed low-light scenes is proposed. We carefully design the scene by controlling the type and power of the light source and generating noisy spike streams based on Zhao et al. (2022a). Besides, we propose a light-robust reconstruction method as shown in Fig. 1(b). Specifically, to compensate for information deficiencies in low-light spike streams, we propose a **l**ight-**r**obust **rep**resentation (LR-Rep). In LR-Rep, the release time of forward and backward spikes is used to update a **g**lobal **i**nter-**s**pike **i**nterval (GISI). Then, to further excavate temporal information in spike streams, LR-Rep is fused with forward (backward) temporal features. During the feature fusion process, we add alignment information to avoid the misalignment of motion from different timestamps. Finally, the scene is clearly reconstructed from fused features.

Empirically, we show the superiority of our reconstruction method. Importantly, our method also generalizes well to real spike streams. In addition, extensive ablation studies demonstrate the effectiveness of each component. The main contributions of this paper can be summarized as follows:

• A reconstruction benchmark for high-speed low-light scenes is proposed. We carefully construct varied low-light scenes that are close to reality.

• We propose a recurrent-based reconstruction framework where a light-robust representation, LR-Rep, and fusion module can effectively compensate for information deficiencies in low-light spike streams.

• Experimental results on real and synthetic datasets have shown our method can more effectively handle spike streams under different light conditions than previous methods.

## 2 RELATED WORK

### 2.1 LOW-LIGHT VISION

Low-light environment has always been a challenge not only for human perception but also for computer vision methods. For traditional cameras, some works concern the enhancement of low-light images. Wei et al. (2018) propose the LOL dataset containing low/normal-light image pairs and propose a deep Retinex-Net including a Decom-Net for decomposition and an Enhance-Net for illumination adjustment. Jiang et al. (2021) proposes the EnlightenGAN which is first trained on unpaired data to low-light image enhancement. Guo et al. (2020) proposes Zero-DCE which formulates light enhancement as a task of image-specific curve estimation with a deep network. Besides, some work focuses on the robustness of vision task to low-light, *e.g.,* object detection. Sasagawa & Nagahara (2020) proposes a method of domain adaptation for merging multiple models to handle objects in a low-light situation. Li et al. (2021a) integrates a new non-local feature

aggregation method and a knowledge distillation technique to with the detector networks. Wang et al. (2023) combines with the image enhancement algorithm to improves the accuracy of object detection. For spike camera, it are also affected by low-light environments. Dong et al. (2022) propose a real low-light high-speed dataset for reconstruction. However, it lacks corresponding image sequences as ground truth. To solve the challenge in the reconstruction of low-light spike streams, We have fully developed the task including a reconstruction benchmark for high-speed low-light scenes and a light-robust reconstruction method.

## 2.2 RECONSTRUCTION FOR SPIKE CAMERA

The reconstruction of high-speed dynamic scenes has been a popular topic for spike camera. Based on the statistical characteristics of spike stream, Zhu et al. (2019) first reconstruct high-speed scenes. Zhao et al. (2020) improved the smoothness of reconstructed scenes by introducing motion aligned filter. Zhu et al. (2022) construct a dynamic neuron extraction model to distinguish the dynamic and static scenes. For enhancing reconstruction results, Zheng et al. (2021) uses short-term plasticity mechanism to exact motion area. Zhao et al. (2021) first proposes a deep learning-based reconstruction framework, Spk2ImgNet (S2I), to handle the challenges brought by both noise and high-speed motion. chen et al. (2022) build a self-supervised reconstruction framework by introducing blind-spot networks. It achieves desirable results compared with S2I. The reconstruction method (Zhang et al., 2023) presents a novel Wavelet Guided Spike Enhancing (WGSE) paradigm. By using multi level wavelet transform, the noise in the reconstructed results can be effectively suppressed.

## 2.3 SPIKE CAMERA SIMULATION

Spike camera simulation is a popular way to generate spike streams and accurate labels. Zhao et al. (2021) first convert interpolated image sequences with high frame rate into spike stream. Based on Zhao et al. (2021), Zhu et al. (2021); Kang et al. (2021); Zhao et al. (2022a) add some random noise to generate spike streams more accurately. To avoid motion artifacts caused by interpolation, Hu et al. (2022) presents the spike camera simulator (SPCS) combining simulation function and rendering engine tightly. Then, based on SPCS, optical flow datasets for spike camera are first proposed. Zhang et al. (2022) generate the first spike-based depth dataset by the spike camera simulation.

## 3 RECONSTRUCTION DATASETS

In order to train and evaluate the performance of reconstruction methods in low-light high-speed scenes, we propose two low-light spike stream datasets, **R**and **L**ow-**L**ight **R**econstruction (RLLR) and **L**ow-**L**ight **R**econstruction (LLR) based on spike camera model. RLLR is used as our train dataset and LLR is carefully designed to evaluate the performance of different reconstruction methods as test dataset. We first introduce the spike camera model, and then introduce our datasets where noisy spike streams are generated by the spike camera model.

**Spike camera model**    Each pixel on the spike camera model converts light signal into current signal and accumulate the input current. For pixel $\mathbf{x} = (x, y)$, if the accumulation of input current reaches a fixed threshold $\phi$, a spike is fired and then the accumulation can be reset as,

$$A(\mathbf{x}, t) = A_{\mathbf{x}}(t) \bmod \phi = \int_0^t I_{tot}(\mathbf{x}, \tau) d\tau \bmod \phi, \tag{1}$$

$$I_{tot}(\mathbf{x}, \tau) = I_{in}(\mathbf{x}, \tau) + I_{dark}(\mathbf{x}, \tau), \tag{2}$$

where $A(\mathbf{x}, t)$ is the accumulation at time $t$, $A_{\mathbf{x}}(t)$ is the accumulation without reset before time $t$, $I_{in}(\mathbf{x}, \tau)$ is the input current at time $\tau$ (proportional to light intensity) and $I_{dark}(\mathbf{x}, \tau)$ is the main fixed pattern noise in spike camera, *i.e.,* dark current (Zhu et al., 2021; Zhao et al., 2022a). Further, due to limitations of circuits, each spike is read out at discrete time $nT, n \in \mathbb{N}$ ($T$ is a micro-second level). Thus, the output of the spike camera is a spatial-temporal binary stream $S$ with $H \times W \times N$ size. The $H$ and $W$ are the height and width of the sensor, respectively, and $N$ is the temporal window size of the spike stream. According to the spike camera model, it is natural that the spikes (or information) in low-light spike streams are sparse because reaching the threshold is lengthy. More details about low-light spike streams are in appendix.

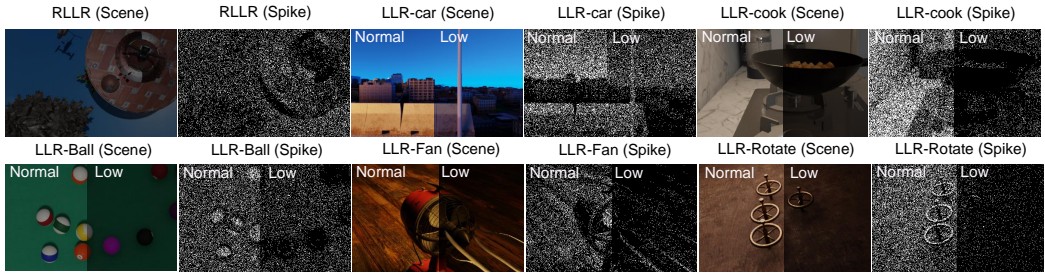

Figure 2: Proposed datasets, RLLR and LLR. RLLR includes the random scenes and LLR includes carefully designed scenes. More details are in our appendix.

**RLLR**  As shown in Fig. 2, RLLR includes 100 random low-light high-speed scenes where high-speed scenes are first generated by SPCS (Hu et al., 2022) and then the light intensity of all pixels in each scene is darkened by multiplying a random constant (0-1). Each scene in RLLR continuously records a spike stream with $400 \times 250 \times 1000$ size and corresponding image sequence. Then, for each image, we clip a spike stream with $400 \times 250 \times 41$ size from the spike stream as input.

**LLR**  As shown in Fig. 2, LLR includes $5 \times 2$ carefully designed high-speed scenes where we use the scenes with five kinds of motion (named Ball, Car, Cook, Fan and Rotate) and each scene corresponds to two light sources (normal and low). To ensure the reliability of our scenes, different light sources are used, and the power of light source is consistent with the real world. Each scene in LLR continuously records 21 spike streams with $400 \times 250 \times 41$ size and 21 corresponding images.

In proposed datasets, we consider noise of spike camera based on Zhao et al. (2022a). We further discuss the impact of noisy (noise-free) spike streams on the performance of methods in the appendix.

## 4 METHOD

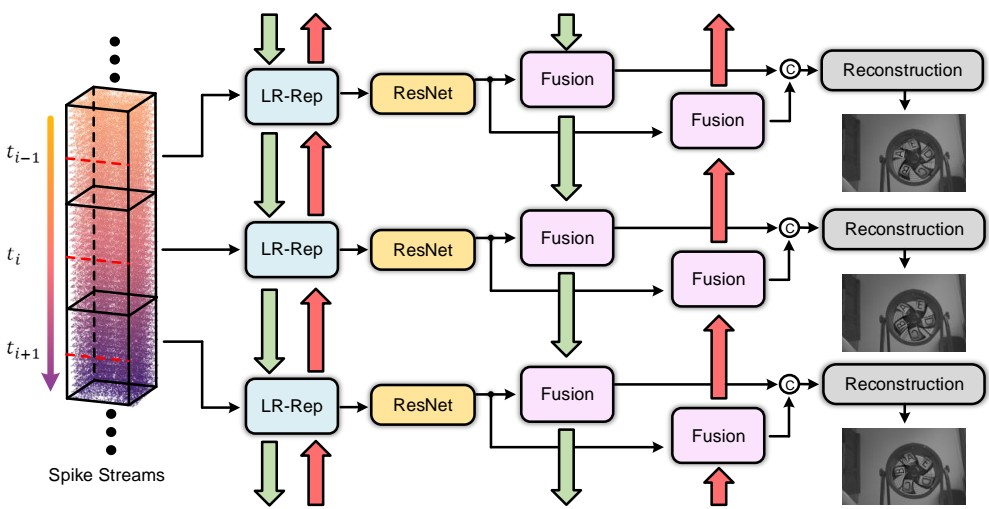

Figure 3: Illustration of the proposed recurrent-based reconstruction framework. It includes a light-robust representation, feature extractor (ResNet), fusion and reconstruction. The green and red lines represent the forward and backward data flow. The two kinds of data flow are independent.

### 4.1 PROBLEM STATEMENT

For simplicity, we write $\mathbf{S}_t \in \{0,1\}^{H \times W \times (2\Delta t+1)}$ to denote a spike stream from time $t - \Delta t$ to $t + \Delta t$ ($2\Delta t + 1$ is the fixed temporal window) and write $\mathbf{Y}_t \in \mathbb{R}^{H \times W}$ to denote the instantaneous

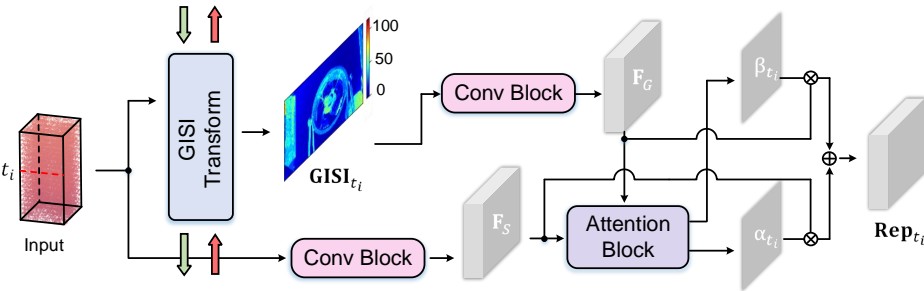

Figure 4: Illustration of the proposed light-robust representation. We use convolution blocks to extract shallow features from input spike stream and GISI, respectively. Then they are fused by an attention block.

light intensity received in spike camera at time $t$. Reconstruction is to use continuous spike streams, $\{\mathbf{S}_{t_i}, t_i = i * (2\Delta t + 1)|i = 1, 2, 3...K\}$ to restore the light intensity information at different time, $\{\mathbf{Y}_{t_i}, t_i = i * (2\Delta t + 1)|i = 1, 2, 3...K\}$. Generally, the temporal window $2\Delta t + 1$ is set as 41 which is the same with Zhao et al. (2021); chen et al. (2022); Zhang et al. (2023).

## 4.2 OVERVIEW

To overcome the challenge of low-light spike streams, *i.e.,* the recorded information is sparse (see Fig.1), we propose a light-robust reconstruction method which can fully utilize temporal information of spike streams. It is beneficial from two modules: 1. A light-robust representation, LR-Rep. 2. A fusion module. As shown in Fig. 3, to recover the light intensity information at time $t_i$, $\mathbf{Y}_{t_i}$, we first calculate the light-robust representation at time $t_i$, written as $\mathbf{Rep}_{t_i}$. Then, we use a ResNet module to extract deep features, $\mathbf{F}_{t_i}$, from $\mathbf{Rep}_{t_i}$. $\mathbf{F}_{t_i}$ is fused with forward (backward) temporal features as $\mathbf{F}_{t_i}^f$ ($\mathbf{F}_{t_i}^b$). Finally, we reconstruct the image at time $t_i$, $\hat{\mathbf{Y}}_{t_i}$ with $\mathbf{F}_{t_i}^f$ and $\mathbf{F}_{t_i}^b$.

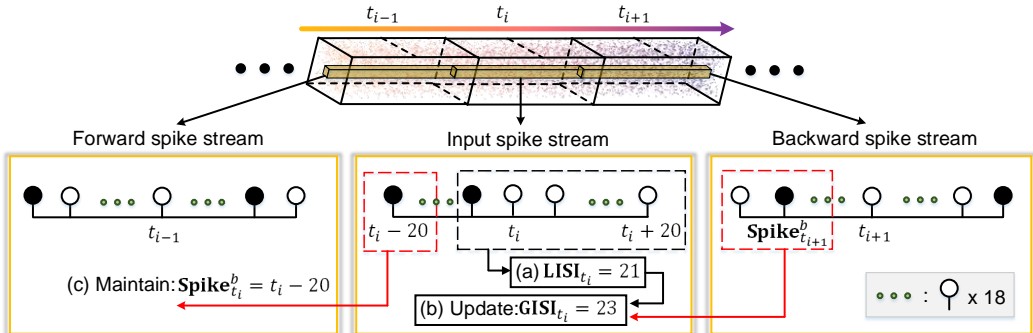

Figure 5: Illustration of GISI transform for backward in a pixel. (a). Calculate the local inter-spike interval, $\mathbf{LISI}_{t_i}$ from the input spike stream (chen et al., 2022; Zhao et al., 2022b). (b). Update global inter-spike interval, $\mathbf{GISI}_{t_i}$ based on the release time of backward spike, $\mathbf{Spike}_{t_{i+1}}^b$ and $\mathbf{LISI}_{t_i}$. (c). Maintain and transmit the release time of backward spike, $\mathbf{Spike}_{t_i}^b$. Black (white) circle is a (no) spike and the red line is backward data flow. More details are shown in algorithm. 1.

## 4.3 LIGHT-ROBUST REPRESENTATION

As shown in Fig. 4, a light-robust representation, LR-Rep, is proposed to aggregate the information in low-light spike streams. LR-Rep mainly consists of two parts, GISI transform and feature extraction.

**GISI transform** Calculating the local inter-spike interval from the input spike stream is a common operation (chen et al., 2022; Zhao et al., 2022b) and we call it as LISI transform. Different from LISI transform, we propose a GISI transform which can utilize the release time of forward and backward

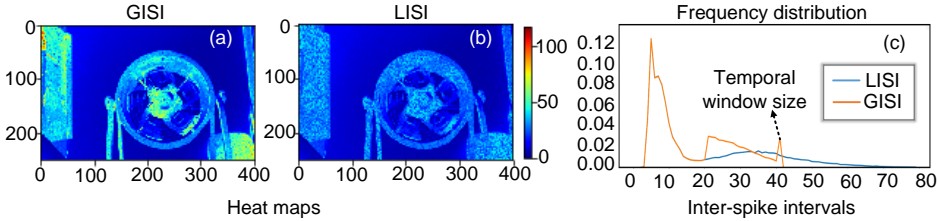

Figure 6: (a) and (b) show the visualizations of $\mathbf{GISI}_{t_i}$ and $\mathbf{LISI}_{t_i}$ in a real spike stream. (c) shows the distribution of pixel-wise values in $\mathbf{GISI}_{t_i}$ and $\mathbf{LISI}_{t_i}$.

spikes to obtain the global inter-spike interval $\mathbf{GISI}_{t_i}$. It need to be performed twice, i.e. once forward and once backward respectively. Taking GISI transform for backward as an example, it can be summarized as three steps as shown in Fig. 5. Our GISI transform can extract more temporal information from spike streams than LISI transform as shown in Fig. 6.

**Feature extraction** After GISI transform, we separately extract shallow features of $\mathbf{GISI}_{t_i}$ and input spike stream, $\mathbf{F}_G$ and $\mathbf{F}_S$ through convolution block. Finally, $\mathbf{Rep}_{t_i}$ is obtained by an attention module where $\mathbf{F}_G$ and $\mathbf{F}_S$ are integrated, *i.e.,*

$$[\beta_{t_i}, \alpha_{t_i}] = Att([\mathbf{F}_G, \mathbf{F}_S]), \tag{3}$$

$$\mathbf{Rep}_{t_i} = \beta_{t_i}\mathbf{F}_G + \alpha_{t_i}\mathbf{F}_S, \tag{4}$$

where $Att(\cdot)$ denotes an attention block including 3-layer convolution with 3-layer activation function and $\mathbf{Rep}_{t_i}$ is our LR-Rep at time $t_i$. Related details are in our appendix.

## 4.4 FUSION AND RECONSTRUCTION

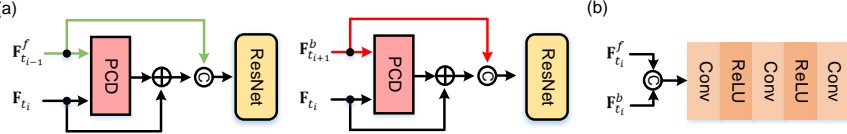

Figure 7: Illustration of fusion module and reconstruction module. (a) denotes forward (green line) and backward (red line) fusion modules. (b) denotes reconstruction module.

We first extract the deep feature $\mathbf{F}_{t_i}$ of $\mathbf{Rep}_{t_i}$ through a ResNet with 16 layers. Then, as shown in Fig. 7(a), for forward, temporal features $\mathbf{F}_{t_{i-1}}^f$ and $\mathbf{F}_{t_i}$ are fused as temporal features of the input spike stream $\mathbf{F}_{t_i}^f$. For backward, temporal features $\mathbf{F}_{t_{i+1}}^b$ and $\mathbf{F}_{t_i}$ are fused as temporal features of the input spike stream $\mathbf{F}_{t_i}^b$. To avoid the misalignment of motion from different timestamps, we use a Pyramid Cascading and Deformable convolution (PCD) (Wang et al., 2019) to add alignment information to $\mathbf{F}_{t_i}$. The above process can be written as,

$$\mathbf{F}_{t_i} = f(\mathbf{Rep}_{t_i}), \tag{5}$$

$$\mathbf{F}_{t_i}^f = f([\mathbf{F}_{t_i} + a(\mathbf{F}_{t_{i-1}}^f, \mathbf{F}_{t_i}), \mathbf{F}_{t_{i-1}}^f]), \tag{6}$$

$$\mathbf{F}_{t_i}^b = f([\mathbf{F}_{t_i} + a(\mathbf{F}_{t_{i+1}}^b, \mathbf{F}_{t_i}), \mathbf{F}_{t_{i+1}}^b]), \tag{7}$$

where $f(\cdot)$ denotes the feature extraction and $a(\cdot, \cdot)$ denotes the PCD module. Finally, as shown in Fig. 7(b), we use forward and backward temporal features ($\mathbf{F}_{t_i}^b$ and $\mathbf{F}_{t_i}^f$) to reconstruct the current scene at time $t_i$, *i.e.,*

$$\hat{\mathbf{Y}}_{t_i} = c([\mathbf{F}_{t_i}^b, \mathbf{F}_{t_i}^f]), \tag{8}$$

$$\mathcal{L} = \sum_{i=1}^{K} \|\hat{\mathbf{Y}}_{t_i} - \mathbf{Y}_{t_i}\|_1 \tag{9}$$

where $c(\cdot)$ denotes 3-layer convolution with 2-layer ReLU, $\mathcal{L}$ is loss function, $\|\cdot\|_1$ denotes 1-norm and $K$ is the number of continuous spike streams.

## 5 EXPERIMENT

### 5.1 IMPLEMENTATION DETAILS

We train our method in the proposed dataset, RLLR. Consistent with previous work Zhao et al. (2021); chen et al. (2022); Zhang et al. (2023), the temporal window of each input spike stream is 41. The spatial resolution of input spike streams is randomly cropped the spike stream to $64 \times 64$ during the training procedure and the batch size is set as 8. Besides, forward (backward) temporal features and the release time of spikes in our method are maintained from 21 continuous spike streams. We use Adam optimizer with $\beta_1 = 0.9$ and $\beta_2 = 0.99$. The learning rate is initially set as 1e-4 and scaled by 0.1 after 70 epochs. The model is trained for 100 epochs on 1 NVIDIA A100-SXM4-80GB GPU.

### 5.2 RESULTS

We compare our method with traditional reconstruction methods, *i.e.,* TFI (Zhu et al., 2019), STP (Zhu et al., 2021), SNM (Zhu et al., 2022) and deep learning-based reconstruction methods, *i.e.,* SSML (chen et al., 2022), Spk2ImgNet (S2I) (Zhao et al., 2021), WGSE (Zhang et al., 2023). The supervised learning methods, S2I and WGSE, are retrained on RLLR and the parameter configuration is the same as their respective papers. We evaluate methods on two kinds of data:
(1) The carefully designed **synthetic** dataset, LLR.
(2) The **real** spike streams dataset, PKU-Spike-High-Speed (Zhao et al., 2021) and low-light high-speed spike streams dataset (Dong et al., 2022).

Table 1: PSNR, PSNR (scale), and SSIM of reconstruction results on synthetic dataset, LLR. PSNR (scale): PSNR is calculated after scaling both the ground truth and reconstructed images to 0-255.

| Metric | TFI | SSML | S2I | STP | SNM | WGSE | Ours |
|---|---|---|---|---|---|---|---|
| PSNR | 31.409 | 38.432 | 40.883 | 24.882 | 25.741 | 42.959 | **45.075** |
| PSNR (scale) | 21.665 | 30.176 | 31.202 | 14.894 | 18.527 | 32.439 | **38.131** |
| SSIM | 0.72312 | 0.89942 | 0.95915 | 0.55537 | 0.80281 | 0.97066 | **0.98681** |

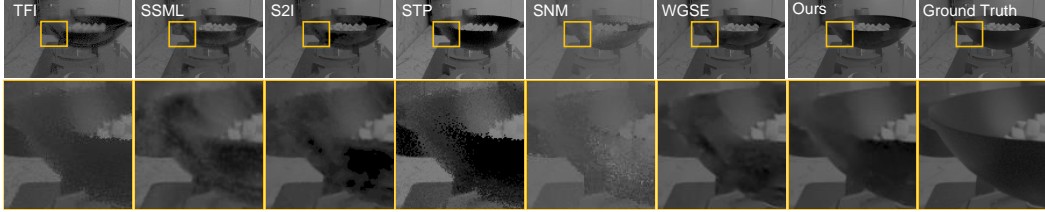

Figure 8: Results from different reconstruction methods on our LLR. More results are in appendix.

**Results on our synthetic dataset** As shown in Table. 1, we use the two reference image quality assessment (IQA) metrics, *i.e.,* PSNR and SSIM to evaluate the performance of different methods on LLR. We can find that our method achieves the best reconstruction performance and has a PSNR gain over 4dB than the reconstruction method, S2I, which demonstrates its effectiveness. Fig. 8 shows the visualization results from different reconstruction methods. We can find that our method can better restore motion details in dark regions than other methods.

**Results on real datasets** For real data, we test different methods on two spike stream datasets, PKU-Spike-High-Speed (Zhao et al., 2021) and low-light spike streams (Dong et al., 2022). PKU-Spike-High-Speed includes 4 high-speed scenes under normal-light conditions and Dong et al. (2022) includes 5 high-speed scenes under low-light conditions. Fig. 9 shows the reconstruction results. Note that we apply the traditional HDR method (Ying et al., 2017) to reconstruction results on Dong et al. (2022) because scenes are too dark. Our method can more effectively restore the information in scenes i.e., clear texture and less noise.

We perform a user study written as US (Wilson, 1981; Jiang et al., 2021) to quantify the visual quality of different methods. For each scene in datasets, we randomly select reconstructed images at the same

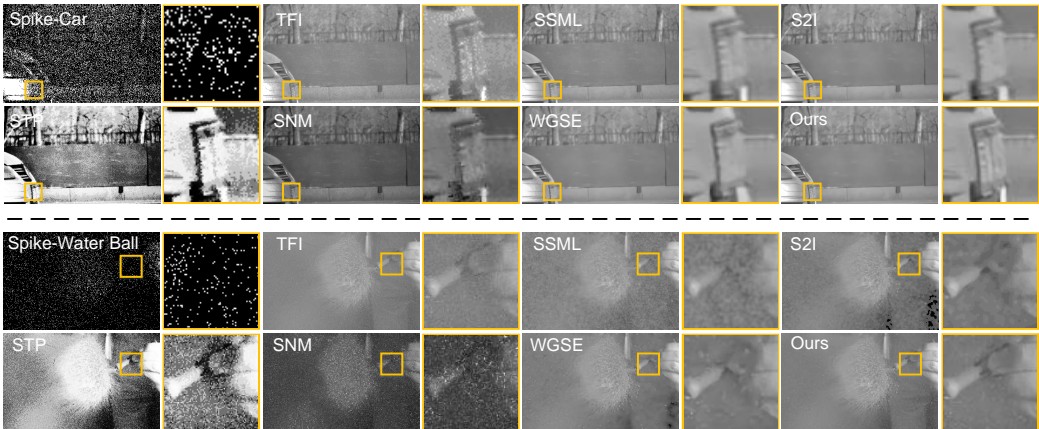

Figure 9: Results from different reconstruction methods on the real datasets, PKU-Spike-High-Speed (Top) and low-light high-speed spike streams dataset (Bottom). More results are in our appendix.

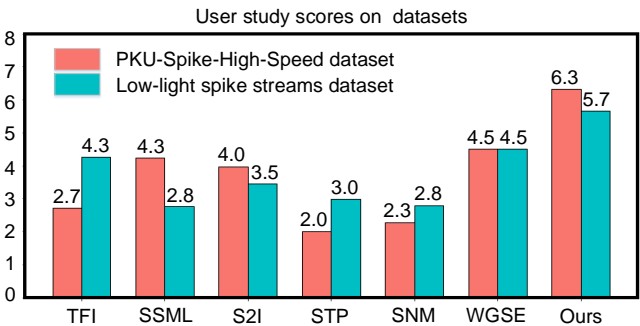

Figure 10: User study scores (↑) of reconstructed images from different methods. The max (min) score is 7 (1).

time from different methods and display them on the screen (the image order is randomly shuffled). 30 human subjects (university degree or above) are invited to independently score the visual quality of the reconstructed image. The scores of visual quality range from 1 to 7 (worst to best quality). The average subjective scores for each spike stream dataset are shown in Fig. 10 and our method reaches the highest US score in all methods.

### 5.3 ABLATION

**Proposed modules**    To investigate the effect of the proposed light-robust representation LR-Rep, the **a**djacent (forward and backward) **d**eep temporal **f**eatures (ADF), *i.e.,* $\mathbf{F}_{t_i}^{b}$ and $\mathbf{F}_{t_i}^{f}$ in our fusion module, the **a**lignment **i**nformation in our **f**usion module (AIF) and GISI transform in LR-Rep, we compare 5 baseline methods with our final method. (A) is the basic baseline without LR-Rep, ADF and AIF. Table. 2 shows ablation results on proposed dataset, LLR. The comparison between (A) and (C) ((B) and (D)) proves the effectiveness of LR-Rep. The comparison between (A) and (B) ((C) and (D)) proves the effectiveness of ADF. Further, by adding the alignment information in the fusion module *i.e.,* AIF, our final method (F) appropriately reduces the misalignment of motion from different timestamps and can reconstruct high-speed scenes more accurately than (D). Besides, the comparison between (E) and (F) shows GISI has better performance than LISI. It is because GISI can extract more temporal information than LISI (see Fig. 6).

**Comparison with other representation**    We compare the performance of different representation in our framework, *i.e.,* (1) General representation of spike stream: TFI and TFP (Zhu et al., 2019) (2) Tailored representation for reconstruction networks: AMIM (chen et al., 2022) in SSML, SALI (Zhao et al., 2021) in S2I and WGSE-1d (Zhang et al., 2023) in WGSE. We replace LR-Rep in our

method as above representation. They are retrained on the dataset, RLLR and implementation details are the same with our method. As shown in Table. 3, our LR-Rep achieves the best performance which means LR-Rep can better adapt to our framework.

Table 2: Abltion results on synthetic dataset, LLR.

| Index | Effect of different network structures | PSNR | SSIM |
|---|---|---|---|
| (A) | Removing LR-Rep & AIF & ADF | 42.743 | 0.97403 |
| (B) | Removing LR-Rep & AIF | 44.151 | 0.98514 |
| (C) | Removing AIF & ADF | 44.739 | 0.98636 |
| (D) | Removing AIF | 44.956 | 0.98678 |
| (E) | Replace GISI with LISI | 44.997 | 0.98676 |
| (F) | Our final method | **45.075** | **0.98681** |

Table 3: Performance of different representation methods in our framework. All methods are trained on RLLR and are tested on LLR.

| Rep. | TFP | TFI | AMIM | SALI | WGSE-1d | LR-Rep |
|---|---|---|---|---|---|---|
| PSNR | 38.615 | 37.617 | 41.95 | 43.314 | 42.302 | **45.075** |
| SSIM | 0.96641 | 0.93632 | 0.97493 | 0.98304 | 0.97438 | **0.98681** |

**The number of continuous spike streams**   For solving the reconstruction difficulty caused by inadequate information in low-light scenes, the release time of spike in LR-Rep and temporal features in fusion module are maintained forward and backward in a recurrent manner. The number of continuous spike streams has an impact on our method performance. Fig. 11 shows its effect on the performance. We can find that, as the number increases, the performance of our method can greatly increase until convergence. This is because, as the number increases, our method can utilize more temporal information until sufficient. The reconstrued image from 21 continuous spike streams has more details in a shaded area.

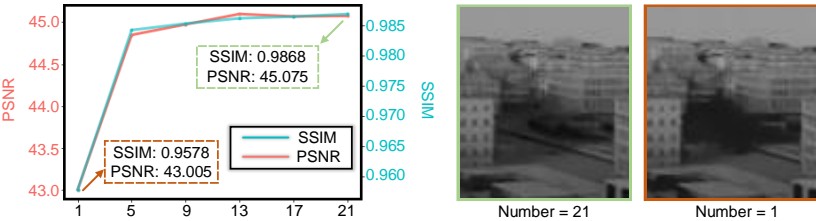

Figure 11: Effect of the number of continuous spike streams on the performance. We test on the dataset, LLR.

## 6   CONCLUSION

We propose a recurrent-based reconstruction framework for spike camera to better handle different light conditions. In our framework, a light-robust representation (LR-Rep) is designed to aggregate temporal information in spike streams. Moreover, a fusion module is used to extract temporal features. To evaluate the performance of different methods in low-light high-speed scenes, we synthesize a reconstruction benchmark where light sources are carefully designed to be consistent with reality. The experiment on both synthetic data and real data shows the superiority of our method.

**Limitation**   Due to the need to fuse both forward and backward temporal features, our method is offline, *i.e.,* After spike camera collects spike streams for a long period of time, the data can be reconstructed. However, real-time applications may require spike stream reconstruction to be performed while the spike camera is capturing the scene. In future work, we plan to extend our method so to online reconstruct.

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

## A  APPENDIX

### A.1  LOW-LIGHT SPIKE STREAMS ANALYSIS

In low-light spike streams, the valid information can greatly decrease. It includes two reasons: (a) The information carried by each spike from the input signal is greatly reduced due to the interference of noise. (b) The total number of spikes in spike streams decrease greatly. Based on (1) in our main paper, we can get the valid accumulation in a spike. First, for the pixel $\mathbf{x}$, the time to fire a spike, $t_{\mathbf{x}}$, can be written as,

$$t_{\mathbf{x}} = A_{\mathbf{x}}^{-1}(\phi). \tag{10}$$

Note that $A_{\mathbf{x}}^{-1}(\cdot)$ exists because $A_{\mathbf{x}}(\cdot)$ is monotonically increasing. Especially, when the light intensity is fixed, $i.e.,$ $I_{in}(\mathbf{x}, \tau) = I$, (11) can be written as,

$$t_{\mathbf{x}} = \phi(I + I_{dark}(\mathbf{x}))^{-1}. \tag{11}$$

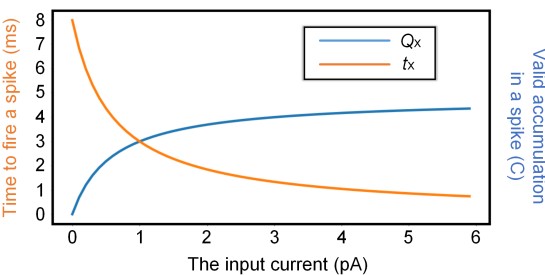

Figure 12: Influence of input current on spikes. In low-light environment, $i.e.,$ input current is low, the time to fire a spike is long and the valid accumulation in each spike is small.

Further, we can get the valid accumulation from input current $I$ in a spike, $Q_{\mathbf{x}}(I)$, as,

$$Q_{\mathbf{x}}(I) = It_{\mathbf{x}} = I\phi(I + I_{dark}(\mathbf{x}))^{-1}. \tag{12}$$

The orange curve in Fig. 12 shows that the valid accumulation $Q_{\mathbf{x}}(I)$ increases with increasing input current $I$ which means each spike in low-light environments is more difficult to record information. The blue curve in Fig. 12 shows that time to fire spikes $t_{\mathbf{x}}$ decreases with increasing input current $I$ which means the total information $i.e.,$ the amount of spikes, in low-light spike stream is sparse. The above two characteristics explains the sparsity of information in low-light spike streams.

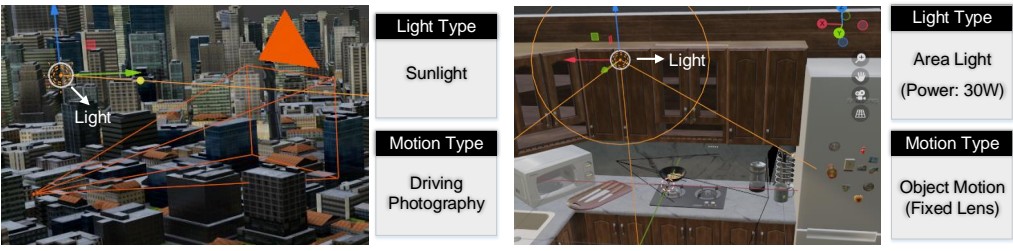

Figure 13: The Car$_N$ (left) and Cook$_L$ (right) in LLR. N (L) means normal (low) light.

### A.2  DATASETS DETAILS

#### A.2.1  SCENE

LLR serves as the test set and is designed to be as consistent as possible with the real world in order to effectively evaluate different methods. To achieve this, as shown in Fig. 13, we have carefully

designed the light source type and the illumination power for each scene to match the real world. Besides, motion of objects is close to the real world. The motion in Ball, Cook, Fan and Rotate is from Hu et al. (2022) while the motion in Car is created based on vehicle speed in real world.

**Light source set** We set the lighting parameters in the advanced 3D graphics software, blender, to make the lighting conditions as consistent as possible with the real world. The following are the configuration details in Blender. In Blender, various types of lighting simulation functions, including sunlight, point lights, and area lights, have been integrated into the graphical interface. We can adjust lighting parameters to control brightness and darkness. For sunlight in Blender, the watts per square meter can be modified. Typically, 100 watts per square meter corresponds to a cloudy lighting environment. For the $Car_L$ scene, we have set sunlight to 10 watts per square meter, which is deemed sufficiently low. For point lights and area lights, Blender allows modification of radiant Power, measured in watts. This is not the electrical power of consumer light bulbs. A light tube with the electrical power of 30W approximately corresponds to a radiant power of 1W. In the $Cook_L$ scene, we have set an area light with the radiant Power to 1W (the electrical power of 30W) . It already represents a very dim indoor light source.

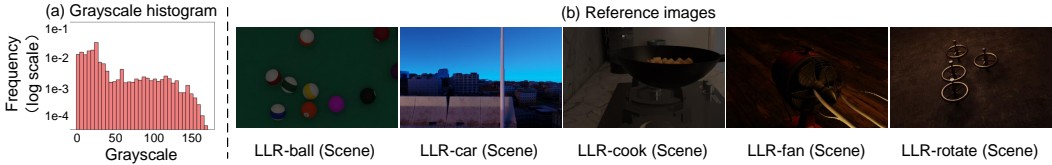

Figure 14: (a) Grayscale histograms of images in low-light scenes, *i.e.,* Ball, Car, Cook, Fan and Rotate with low-light light source. Each bar represents 5 grayscale levels. (b) Reference images.

**Grayscale** The brightness is not only determined by the light source, but also by factors such as camera distance, object occlusion, and so on. These factors are ultimately reflected in the grayscale of the rendered images. Therefore, we calculate the grayscale histograms of images in low-light scenes. As shown in Fig. 14, we can see that the grayscale is diverse and in a lower range. To further demonstrate the performance advantages of our method under different lighting conditions, based on the scene Car, we generate spike streams by modifying the light source parameters. All results are shown in Table 4.

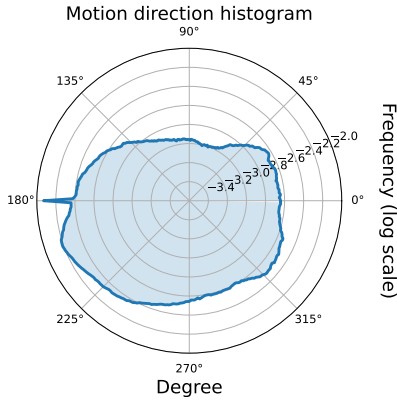

Figure 15: Motion direction histogram of optical flow in LLR.

**Motion** The motion in LLR is diverse. We generate a optical flow every 40 frames for LLR. The degree distribution of the optical flow is in Fig. 15. We can find that the motion in LLR covers all kinds of directions.

Table 4: PSNR and SSIM of reconstruction results under different light sources. We set sunlight in the scene Car to 30, 50, 70, 90, 110 and 130 watts per square meter to render images respectively.

| Light level | 30W | 50W | 70W | 90W | 110W | 130W | Avg. |
|---|---|---|---|---|---|---|---|
| WGSE(PSNR) | 37.12 | 35.68 | 34.45 | 33.88 | 33.61 | 33.36 | 34.68 |
| Ours(PSNR) | **41.52** | **40.69** | **39.99** | **39.51** | **39.15** | **38.91** | **39.96** |
| WGSE(SSIM) | 0.9514 | 0.9418 | 0.9324 | 0.9306 | 0.9305 | 0.9307 | 0.9362 |
| Ours(SSIM) | **0.9772** | **0.9748** | **0.973** | **0.9711** | **0.9701** | **0.9693** | **0.9725** |

Table 5: Reconstruction results on synthetic dataset, LLR. Retrain$_{idea}$: our method is retrained on the noise-free version of RLLR.

| Metric | Our | Retrain$_{idea}$ |
|---|---|---|
| PSNR | **45.075** | 37.679 |
| SSIM | **0.98681** | 0.85374 |

### A.2.2 IMPACT OF SPIKE CAMERA NOISE ON PERFORMANCE

In proposed datasets, we have considered noise of spike camera refer to Zhao et al. (2022a). We further discuss the impact of noisy and noise-free spike streams on the performance of our method as shown in Table. 5. We use an ideal spike camera model in SPCS (Hu et al., 2022) to synthesize a noise-free version of RLLR and retrain our method using the dataset (written as Retrain$_{idea}$). We can find that our method has better performance than Retrain$_{idea}$. Besides, Fig. 16 shows our method can handle noise in real spike streams better than Retrain$_{idea}$.

### A.2.3 TRAIN DATASET SIZE.

The size of train datasets has an impact on the performance of our network. A larger train dataset typically provides more samples and a wider range of variations. In fact, proposed RLLR is enough for the reconstruction task of low-light spike streams. As shown in Table. 6, we find that as the dataset size increase, the performance of the model also improve. However, it is observed that the improvement in performance becomes less significant after the dataset size reaches 60% of RLLR. It shows that the proposed RLLR is sufficient for training our network.

## A.3 LR-REP DETAILS

### A.3.1 GISI TRANSFORM

As shown in Fig. 4 in our main paper, we first use the GISI transform to get the global inter-spike interval, $\mathbf{GISI}_{t_i}$, from the input spike stream and the release time of forward and backward spikes. The GISI transform can be summarized as three steps (see Fig. 5): (a). Calculate the local inter-spike interval from input spike stream as chen et al. (2022); Zhao et al. (2022b) and we call it LISI transform for simplicity. (b). Update the local inter-spike interval as global inter-spike interval based on the release time of forward and backward spikes. (c). Maintain the release time of forward (backward) spikes of backward (forward) spike streams. Related details are shown in Algorithm.1. As shown in Fig. 17, GISI (our final method) not only outperform LISI (Baseline (E) in Table 2) in both PSNR

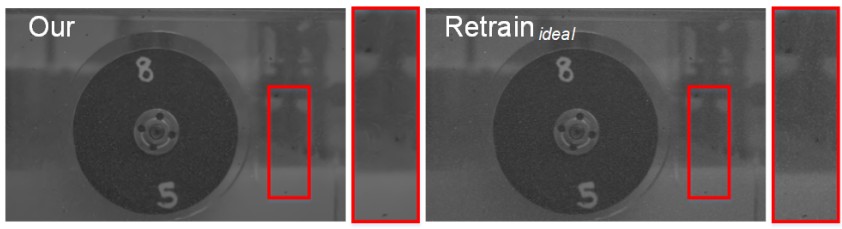

Figure 16: Reconstruction results on a real spike stream. Please enlarge the figure for more details.

Table 6: Evaluation results on LLR. We retrain our network where 20%, 40%, 60%, and 80% of RLLR data are used as training set respectively.

| Metric | 20% | 40% | 60% | 80% | 100% |
|---|---|---|---|---|---|
| PSNR | 35.001 | 38.618 | 44.415 | 44.753 | **45.075** |
| SSIM | 0.93411 | 0.97113 | 0.98459 | 0.98581 | **0.98681** |

and SSIM on LLR but also have better generalization on real data. More importantly, the cost of using GISI instead of LISI is negligible (we only need to use two 400×250 matrices to store the time of the forward spike and the backward spike, respectively), which does not affect the parameter and efficiency of the network.

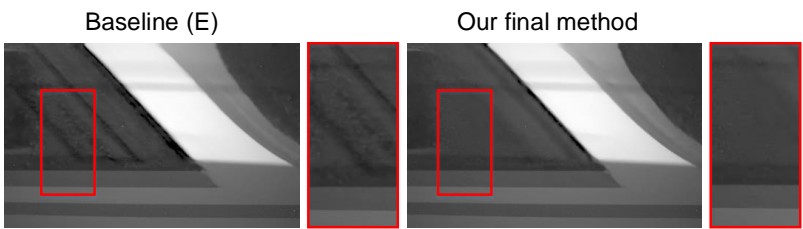

Figure 17: Reconstruction results on a real spike stream. The scene is a high-speed train that exceeds 200 km/h. The glass in the former result (left) shows obvious artifacts, and our result (right) is very smooth and natural.

---

**Algorithm 1** GISI Transform.

**Require:** The spike streams at different time $\{S_{t_i} \mid i = 1, 2, \ldots, K\}$, $K$ is the number of Continuous spike streams.

1: Initialize forward state $\text{Spike}_{t_1}^{forward} = 0$.
2: Initialize backward state $\text{Spike}_{t_K}^{backward} = 2K\Delta t$.
3: **for** $i$ from 1 to $K$ **do**
4:      Calculate $\text{LISI}_{t_i}$ based on $S_{t_i}$.
5: **end for**
6: **for** $i$ from 2 to $K$ **do**
7:      Forward search the recent release time of spike to $t_{i+1}$, $\text{Spike}_{t_i}^{forward}$ based on $S_{t_i}$.
8:      **if** $\text{Spike}_{t_i}^{forward}$ is None **then**
9:          Set $\text{Spike}_{t_i}^{forward} = \text{Spike}_{t_{i-1}}^{forward}$.
10:      **end if**
11:      Update $\text{GISI}_{t_i}$ based on $S_{t_i}$ and $\text{Spike}_{t_{i-1}}^{forward}$.
12: **end for**
13: **for** $i$ from $K-1$ to 1 **do**
14:      Backward search the recent release time of spike to $t_{i-1}$, $\text{Spike}_{t_i}^{backward}$ based on $S_{t_i}$.
15:      **if** $\text{Spike}_{t_i}^{backward}$ is None **then**
16:          Set $\text{Spike}_{t_i}^{backward} = \text{Spike}_{t_{i+1}}^{backward}$.
17:      **end if**
18:      Update $\text{GISI}_{t_i}$ based on $S_{t_i}$ and $\text{Spike}_{t_{i+1}}^{backward}$.
19: **end for**
20: Return $\{\text{GISI}_{t_i} \mid i = 1, 2, \ldots, K\}$

---

### A.3.2 ROBUSTNESS TO LIGHT CONDITION

In LR-Rep, we utilize an attention mechanism to fuse the input spike stream and proposed global inter-spike interval (GISI) to extract shallow features of areas with different brightness. We first state

the fact that under normal-light condition, the input spike stream contains sufficient information and can be well extracted by the network Zhao et al. (2021); chen et al. (2022). Under low-light condition, proposed GISI can supplement the missing information in the input spike stream by maintaining the release time of both forward and backward spikes. If we do not maintain the release time of forward and backward spikes, we will obtain local inter-spike intervals (LISI), which will result in the loss of low-light scene information (see Fig. 6). Further, the attention module can adaptively select feature information from both the input spike stream and GISI based on light condition and LR-Rep is light-robust.

Table 7: Comparison between Spk2ImgNet (S2I), WGSE and our method. The input spike stream size is $21 \times 41 \times 250 \times 400$. We test the average of 50 rounds for Inference time.

| Method | Para. | Train time | Inference time |
|--------|-------|------------|----------------|
| S2I | 3.91m | 2h | 1458.45ms |
| WGSE | **3.63m** | **1h** | 1344.06ms |
| Our | 5.32m | 17h | **818.03ms** |

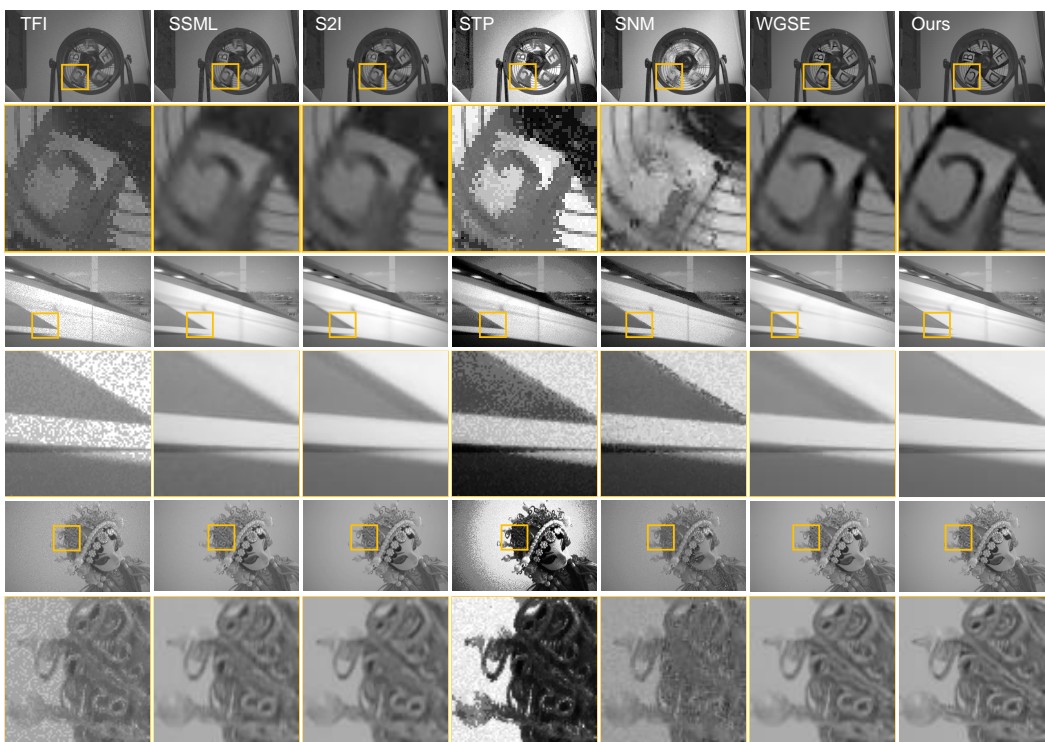

Figure 18: The reconstructed results on the real dataset Zhao et al. (2021).

## A.4 EXPERIMENT

### A.4.1 MODEL EFFICIENCY

Table. 7 demonstrates the training time and inference time of the supervised methods, i.e., Spk2ImgNet, WGSE and our method. Although our method requires more training time compared to Spk2ImgNet and WGSE (Recurrent-based networks typically consume more time during training due to Backpropagation Through Time (BPTT)), our method outperforms Spk2ImgNet and WGSE in terms of inference speed. Besides, due to the need to fuse both forward and backward

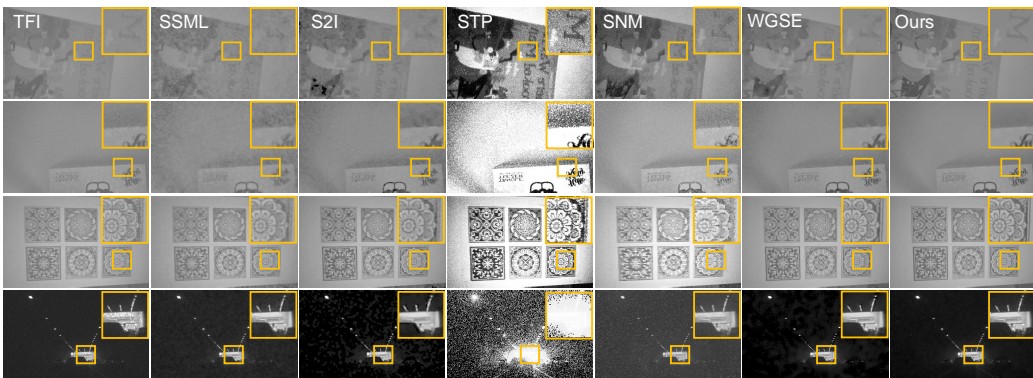

Figure 19: The reconstructed results on the real dataset Dong et al. (2022).

temporal features, our method is offline, i.e., After spike camera collects spike stream for a long period of time, the data can be reconstructed. In future work, we would extend our method so to online reconstruct.

### A.4.2 REAL DATA

Here, we show more results on two real datasets. Fig. 18 and Fig. 19 show more reconstructed images. We find that for traditional methods, TFI performs better on low-light data than TFP, SNM and TFSTP. For deep learning-based methods, SSML introduces a large amount of motion blur while Spk2ImgNet and WGSE may introduces some loss in dark backgrounds. Our method restores texture details in low-light scenes clearly more than other methods. Besides, We also provide the adjusted results from STP based on our reconstruction results as shown in Fig. 20.

### A.4.3 SYNTHETIC DATA

Here, we show more results on synthetic dataset LLR. Fig. 21 shows more reconstruction results on proposed dataset LLR. We find that for traditional methods, TFI performs better on low-light data than TFP, SNM and TFSTP. For deep learning-based methods, SSML introduces a large amount of motion blur while Spk2ImgNet and WGSE may introduces some loss in dark backgrounds. Our method restores texture details in low-light scenes clearly more than other methods.

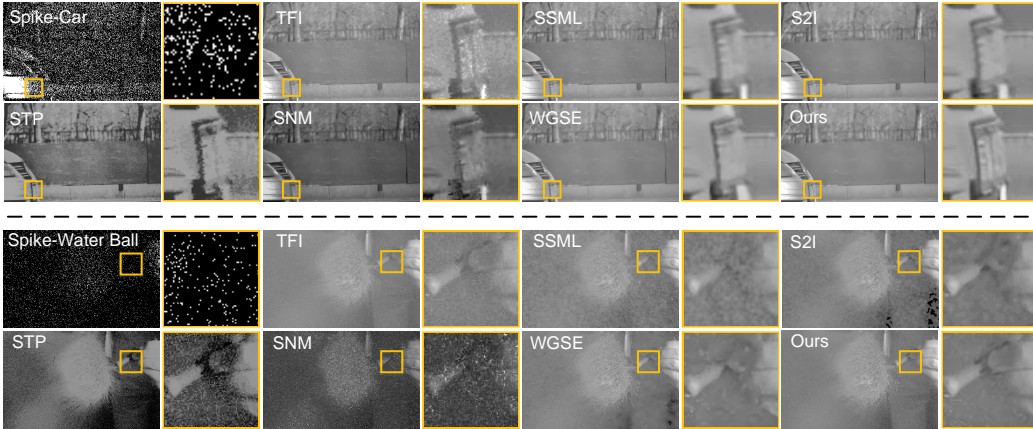

Figure 20: The reconstructed results on the real spike streams. Results from STP are adjusted based on our reconstruction results.

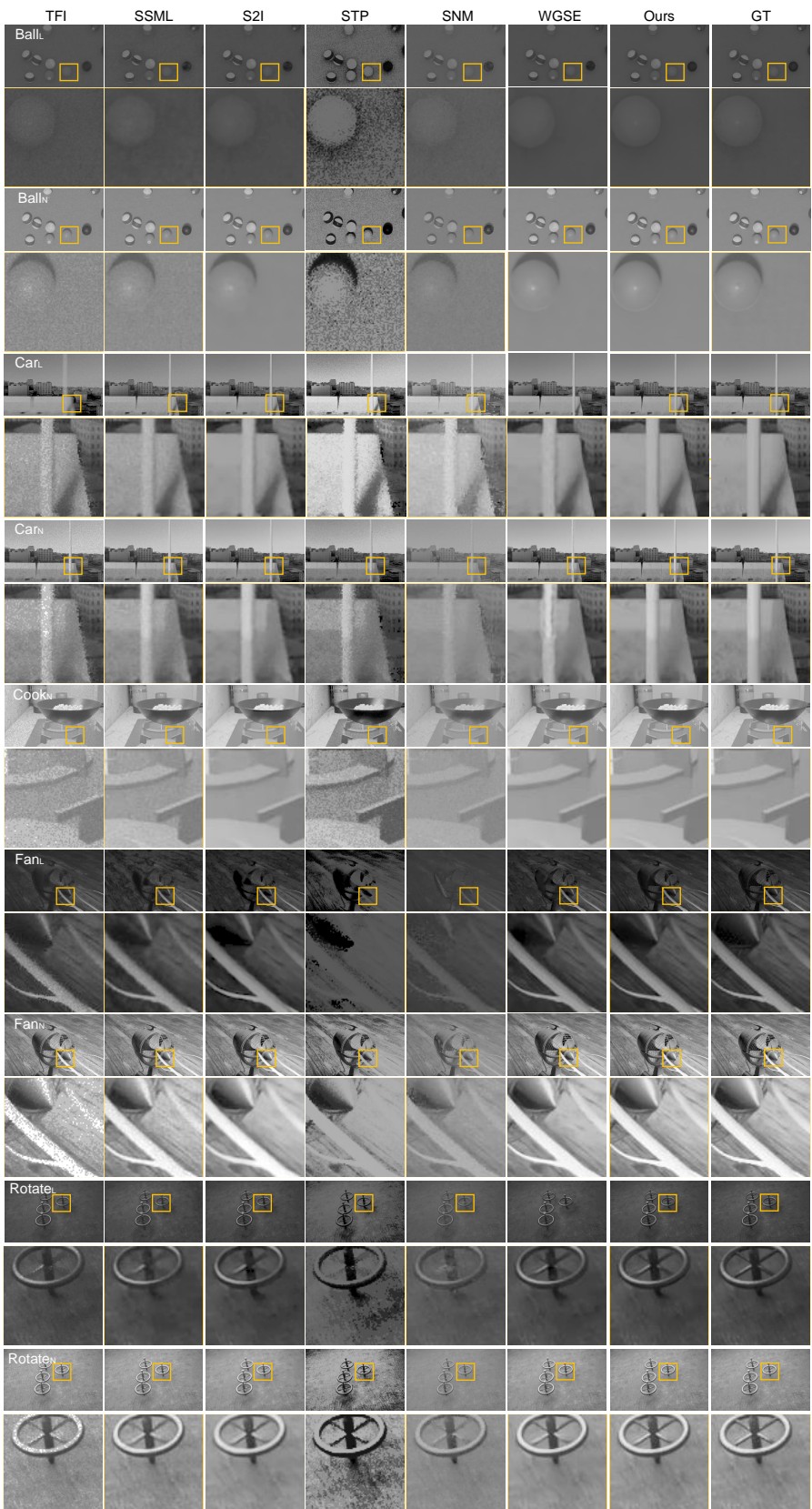

Figure 21: The reconstructed results on LLR. N (L) means normal (low) light.

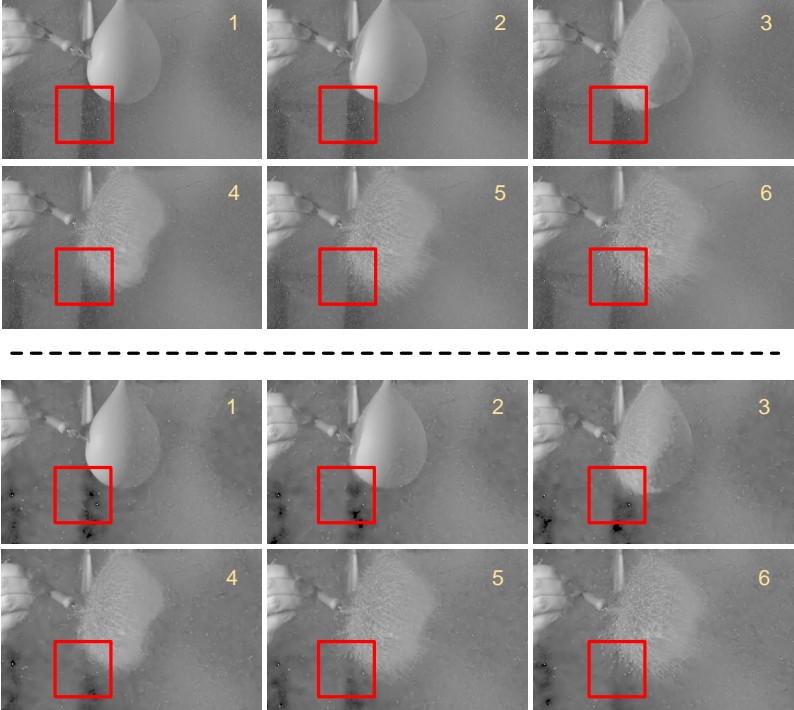

Figure 22: A water polo bursting at high speed in a low-light indoor. We selected the reconstruction results under 6 sampling moments, and the interval between two adjacent sampling moments is 41/40000 s. The above is our method and the bottom is the state-of-the-art reconstruction method Zhang et al. (2023). We apply the traditional HDR method Ying et al. (2017) to reconstruction results because the scene is too dark.

### A.4.4 STABILITY OF CONTINUOUSLY RECONSTRUCTING

Our reconstruction method is stable to spike stream at different moments. Fig. 22 shows the continuous motion of an object in a real high-speed low-light scene. We find that our method can clearly recover motion details at different moments, while the loss introduced by the state-of-the-art WGSE is varied at different time.

### A.4.5 COMPARISON OF QUANTA IMAGE SENSOR

We would like to discuss Quanta Image Sensors (QIS). Spike camera Zhu et al. (2019) and QIS Ma et al. (2022) (including CIS-QIS and SPAD-QIS) share some similar characteristics, such as high temporal resolution and 1-bit (0 or 1) data. Besides, they also have differences in principles and circuits. For one sampling (one frame), QIS records whether a photon has arrived during the sampling, with a corresponding pixel output of 1 if photons arrive, and 0 otherwise Ma et al. (2020). Different from QIS, spike camera continuously accumulates photons Zhu et al. (2019), and if the accumulated value reaches a fixed threshold, the pixel outputs 1 and the accumulation is reset. Otherwise, it outputs 0, and the accumulation value. The different principles result in distinct meanings of two data (QIS data and spike streams). In QIS, 1 reflects the information of a specific sampling. In contrast, in spike camera, 1 contains the information from previous multiple sampling, and adjacent spikes are interdependent. This also leads to differences in the data patterns. This characteristic brings both advantages and disadvantages. In terms of advantages, in spike cameras, the influence of photon shot noise on each spike is reduced as multiple samples of photons are dynamically accumulated together, while QIS is sensitive to poisson shot noise Ma et al. (2022). In terms of disadvantages, spike cameras face more challenges in low-light conditions due to difficulties in reaching the accumulation threshold (see limitation in Zhao et al. (2022b)). Furthermore, the pixel circuits of two cameras are also different. A spike camera continuously accumulates photons in the form of voltage and the voltage can be kept for next sampling. QIS cameras (using SPAD-QIS as an example) amplify the signal through the

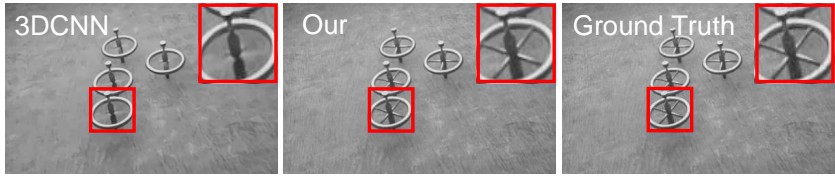

Figure 23: The reconstructed results of Rotate$_N$. Our method has clearer images.

avalanche multiplication mechanism to detect the presence or absence of individual photons Qian et al. (2023). Besides, we test 3DCNN Chandramouli et al. (2019) (a reconstruction method for QIS). To ensure fairness, we retrain 3DCNN using RLLR with spike streams as inputs. Table. 8 demonstrates the reconstruction evaluation on LLR. As shown in Fig. 23, our method removes motion blur better.

Table 8: Reconstruction results on synthetic dataset, LLR. We compare the open source Single Photon Avalanche Diode method, 3DCNN Chandramouli et al. (2019) (ICCP 2019) which is retrained on RLLR.

| Method | PSNR | SSIM |
|--------|--------|---------|
| Our | **45.075** | **0.98681** |
| 3DCNN | 34.507 | 0.93506 |

