# OpenReview forum: "A Light-robust Reconstruction Method for Spike Camera"
_ICLR.cc/2024/Conference — Submitted to ICLR 2024_

### Official Review · Reviewer_bgSj · 2023-10-29

**Soundness:** 2 fair
**Presentation:** 1 poor
**Contribution:** 3 good
**Rating:** 5
**Confidence:** 4

**Summary:**

The authors present a novel reconstruction method for spike cameras specifically in low-light environments. They propose a recurrent-based reconstruction framework that utilizes a light-robust representation (LR-Rep) to aggregate temporal information in spike streams. Additionally, a fusion module is used to extract temporal features. Experimental results on both synthetic and real datasets demonstrate the superiority of their approach. The authors also provide a detailed analysis of low-light spike streams and discuss the efficiency and stability of their method.

**Strengths:**

- Sound logic for problem statement to pipeline development.
- Experiments are well designed, and the performance seems good.

**Weaknesses:**

Generally well written with sound pipeline, there are some points to make improvements. Please check questions.

**Questions:**

- How does the delta T, which sets the length of temporal window affects the overall performance?
- Does the number of forward path of LR-Rep reduces if we increase delta T (i.e., the length of $S_t$ increases)?
- Current pipeline seems to occupy large memory. How much the training takes for the memory and time?

------------------
Related to display or Minor comments:

- Figure 1 Middle(b) is quite confusing. Does the temporal features for both blue and purple are same? If it is different, it should be illustrated differently. Also, the color difference blue and purple are too small, and also it does not looks as purple. Please make it more distinctive. Also, blue and purple arrows seems not an arrows, but justlines, so please improve it for better clarity. Same for the Fig. 3.
- Explanation of LISI transform is in the caption, not in the section. In section 4.3, GISI and LISI suddenly appears, which make readers bit confused. If these are also one of the new blocks proposed, they should be described well.
- In Figure 4, example image of GISI does not show anything. While the figure can be illustrative, suggest authors to use better image that can get the clue what the GISI output looks like.
- In Figure 5, notation is confusing. For example, if the (a) LISI ti=21 mean the LISI output at ti equals 21, "=21" need to be subscript along with ti. Currently it is $LISI_{ti} = 21$. Also suggest to use Latex outputs in the figures for math expressions.
- In Figure 9, maybe the dynamic range are uniform for all reconstructions. The STP shows extremely different range, which makes other reconstruction methods totally not visible. (I expect STP images are totally saturated if we use dynamic range of other images.) How about using two different dynamic ranges, one only for STP and the other for other images. It would be okay with descriptions and specifying the min-max values. Anyway, currently the contrast is too low for reconstructed images.
- In the last line of page 4, typo "camrea".

---

> ### Author Response · Authors · 2023-11-20
>
> Thanks for your careful reading and valuable feedback!
>
> **Q1:  How does the delta T affect the overall performance?**
>
> Our recurrent-based reconstruction method demonstrates flexibility in efficiently covering and extracting temporal information from spike streams. The number of spike streams, combined with delta T, determines the method's coverage of temporal information from spike streams. When employing the same number of spike streams, a too-small delta T results in insufficient coverage of the temporal information. This hinders our method of predicting and reconstructing scenes effectively.  The increasing delta T is beneficial to our method of reconstructing scenes until stable. Generally, delta T is 41  (Zhao et al. ,2021); chen et al. ,2022); Zhang et al. ,2023). We have also opted for this value.
>
>
> **Q2:  Does the number of forward path of LR-Rep reduces if we increase delta T?**
>
> The number of spike streams and delta T are independent.  We can choose the different number of spike streams for inference as shown in Fig.11.
>
> **Q3: The current pipeline seems to occupy a large memory. How much the training takes for memory and time?**
>
>
> The time and memory required for training our method are, 18 hours and 34.4 GB which can be satisfied with 1 NVIDIA A100 GPU. This is attributed to the fact that recurrent-based networks typically consume more time and memory during training due to Backpropagation Through Time (BPTT). During testing, our method demonstrates high efficiency, i.e., our inference time is 818 ms while the inference time of WGSE is 1344 ms as shown in Table 7.
>
>
> **Q4: Related to display**
>
> Thanks for your careful reading. We have made relevant modifications based on your suggestions:
>
>
> 1. Figure 1 \& Figure 3:
> We selected a color pair (green and red) with high contrast to represent forward and backward data streams. We have added a description of that temporary features for both forward and back are different.
>
> 2. Explanation of LISI and GISI transform: We divided the original paragraph into two blocks to describe proposed method more clearly.
>
> 3. Figure 4 \& Figure 3:
> We have selected a more intuitive result (GISI$_t$) for display in **Fig.4**. The reconstruction results in **Fig.3** are also updated accordingly.
>
>
> 4. Figure 5: We have made standardized adjustments to the formula (Not limited to **Fig.5**).
>
> 5. Figure 9: Thanks for your suggestion. The official source code of STP has been used for testing which is comparison a fair comparison. Therefore, we still hope to keep the initial results in the main paper. Besides, we have also provided the adjusted results of STP in **Fig.20**.
>
>
>
>
> 6. typo "camrea": We have updated the relevant typo.

---

> ### Author Response · Authors · 2023-11-21
>
> Thank you for reading! We responded later than expected due to the substantial content additions. However, with the deadline approaching, we sincerely hope all reviewers to respond, allowing us to continue providing further information.

---

> ### Author Response · Authors · 2023-11-23
>
> Thank you for taking the time to read. We have made the necessary modifications based on your suggestions. Additionally, we have provided explanations regarding the questions raised by the public reviewers. We would greatly appreciate your feedback.

---

> ### Author Response · Authors · 2023-11-23
>
> We appreciate the professionalism and fairness of the official reviewers. In the process of responding to your questions, we have also gained insights on how to improve our paper. We sincerely hope that all the reviewers will make a fair and unbiased decision regarding the outcome of this article. If any of the public reviewers can disregard the results presented in the paper, the images and videos in the supplementary material, or even the promised release of model code and datasets, the quality of the open review environment would be greatly diminished.

---

### Official Review · Reviewer_ajbr · 2023-11-01

**Soundness:** 3 good
**Presentation:** 2 fair
**Contribution:** 3 good
**Rating:** 8
**Confidence:** 3

**Summary:**

The paper proposes a recurrent neural network based image reconstruction method for spike cameras.

**Strengths:**

The paper proposes a recurrent neural network based image reconstruction method for spike cameras. The authors generate a synthetic dataset for evaluation. The method is demonstrated to work on real as well as synthetic data.

**Weaknesses:**

1. In Fig. 2, the GT scenes seem to be RGB. How are they converted to gray scale?
2. Why is the PSNR value so high? For eg. the S2I image looks so noisy compared to GT in Fig.1, but according to table 1, the PSNR is > 40dB, which does not make sense.

**Questions:**

Check weaknesses

---

> ### Author Response · Authors · 2023-11-20
>
> Thanks for your careful reading and valuable feedback!
>
> **Q1:  In Fig. 2, the GT scenes seem to be RGB. How are they converted to grayscale?**
>
> We employ the grayscale conversion function provided by OpenCV, i.e., cv2.cvtColor(img, cv2.COLOR\_BGR2GRAY).
>
> **Q2:  Why is the PSNR value so high?  The S2I image looks so noisy compared to GT in Fig.8, but according to table 1, the PSNR is higher than 40dB.**
>
> When the scene is so dark, both the Ground Truth (GT) and the reconstructed images can be at a lower grayscale level. This results in a significantly lower MSE compared to normal-light scenes. Consequently, their PSNR is high.
>
>
> The Peak Signal-to-Noise Ratio (PSNR) can be expressed as $PSNR = 10 \cdot \log_{10}\left(\frac{{\text{255}^2}}{{\text{MSE}}}\right),$
> where MSE represents the Mean Squared Error. It can be further defined as,
> $MSE = \frac{1}{mn} \sum_{i=0}^{m-1} \sum_{j=0}^{n-1} \left( I(i, j) - K(i, j) \right)^2,$where m (n) is the height (width) of the image, respectively and $I(i, j)$($K(i, j)$) represents the pixel value at (x, y) in ground truth (the reconstructed image). Hence, when both the Ground Truth (GT) and the reconstructed images are at a lower grayscale level, MSE is lower (PSNR is higher).  In Fig.8, the scene is exceptionally dark, and we applied the same gamma correction to the reconstructed result for better visibility to readers.

---

> > ### Comment · Reviewer_ajbr · 2023-11-21
> >
> > Q2. The dataset is synthetic. Wouldn't it be better to scale the image to [0,255] after simulation before calculating PSNR? Because the numerator in the "peak" signal to noise ratio (PSNR) is the maximum range of values the image can take. If the image is so dark that it can take values only in [0,10] it does not make sense to keep the numerator as 255.

---

> ### Author Response · Authors · 2023-11-21
>
> Thanks for your positive feedback.  We calculate PSNR after scale the images to [0,255]. The result is as follows:
> ***
> | Metric         | TFI     | SSML    | S2I     | STP     | SNM     | WGSE    | Ours    |
> | -------------- | ------- | ------- | ------- | ------- | ------- | ------- | ------- |
> | PSNR           | 31.409  | 38.432  | 40.883  | 24.882  | 25.741  | 42.959  | **45.075** |
> | PSNR (scale)   | 21.665  | 30.176  | 31.202  | 14.894  | 18.527  | 32.439  | **38.131** |
> | SSIM           | 0.72312 | 0.89942 | 0.95915 | 0.55537 | 0.80281 | 0.97066 | **0.98681** |
>
> ***
> We have updated it to **Table 1**. We also keep the PSNR of raw results in our main paper for two reasons: 1. Although LLR includes low-light scenes, their the dynamic range is wide as shown in **Fig.14**.  2. In previous work (WGSE, SSML and S2I),  PSNR of raw results is shown and we hope to be consistent with those papers.

---

> ### Author Response · Authors · 2023-11-23
>
> We appreciate the professionalism and fairness of the official reviewers. In the process of responding to your questions, we have also gained insights on how to improve our paper. We sincerely hope that all the reviewers will make a fair and unbiased decision regarding the outcome of this article. If any of the public reviewers can disregard the results presented in the paper, the images and videos in the supplementary material, or even the promised release of model code and datasets, the quality of the open review environment would be greatly diminished.

---

### Official Review · Reviewer_sPKH · 2023-11-02

**Soundness:** 3 good
**Presentation:** 2 fair
**Contribution:** 3 good
**Rating:** 6
**Confidence:** 3

**Summary:**

This paper develops and trains a nueral network architecture that can recsontruct high-speed grayscale video frames from the sparse asynchronous data produced by a neuromorphic spiking camera (a camera where each pixel triggers asynchronously at a rate proportional to the intensity of the incident light). The proposed method consists of three steps: It first forms a learned light robust representation of the incoming datastream (pass the incoming data through some convolutional and attention layers). It then passes this representation (combined with features forward and back in time) through a res-net to extract higher-level features. It finally decodes this data into a grayscale video stream.

The proposed algorithm was evaluated on real and experimental low-light spiking camear data. The proposed method slightly but noticeably outperforms the state-of-the-art (to my knowledge) WGSE algorithm.

An ablation study is performed to validate the architectural choices.

**Strengths:**

Lowlight imaging with neuromorphic cameras is an interesting and important problem.

The proposed method noticeably outperforms the state-of-the-art.

The paper includes extensive ablation studies and validation.

**Weaknesses:**

The forward model presented in the main paper doesn't model noise (except quantization). It's unclear how the proposed method would perform in a photon starved regime where a significant amount of Poisson noise would be present.

The paper doesn't link to any reconstructed videos. I can't evaluate if the proposed method introduced significant flickering artifacts.

Figure 5 isn't particularly informative.

Rather than stating, "The algorithm is in appendix", please state where in the appendix the algorithm can be found.

There are a number of typos (e.g., extra capitalizations) that a spell-checker should be able to catch and a few incomplete sentences (e.g., "A fusion module.").

**Questions:**

How would the algorithm behave in the presence of significant Poisson noise?

What were the intuitions behind selecting the chosen architecture? Why not leverage any of the wavelet structure from WGSE?

---

> ### Author Response · Authors · 2023-11-20
>
> Thanks for your careful reading and valuable feedback!
>
> **Q1: Reconstructed videos**
>
> We add reconstructed videos to **the supplementary material** where the state-of-the-art method WGSE is compared. We can find that the results of WGSE are very unstable and fluctuate greatly in the time domain, while our method can effectively handle temporal noise.
>
>
> **Q2: How the proposed method would perform where a significant amount of Poisson noise would be present?**
>
> Our method can deal with Poisson noise better than the state-of-the-art methods. First, the mean of the error from Poisson noise in the temporal dimension is 0. Our bidirectional recurrent-based reconstruction method can effectively aggregate temporal information. Its structure is suitable for learning to filter Poisson noise. Besides, our training set is generated from the spike camera simulator  (Zhao et al.
> 2022a) and it considers Poisson noise and fixed-pattern noise in the spike camera.
>
> **Q3: Figure 5 \& typos**
>
> Thanks for your suggestions. We aim to enhance the reader's understanding of GISI through Fig.5. Following your advice, we have included links to the algorithm. Additionally, we have addressed typos in our paper.
>
> **Q4: What were the intuitions behind selecting the chosen architecture?**
>
> This is an excellent question. We initially observed a pronounced performance degradation of reconstruction methods in low-light conditions. Subsequently, we noticed a significantly sparser distribution of spikes in low-light spike streams compared to those in normal-lighting conditions. The recurrent-based structure proves effective in aggregating information from different time. This forms the basis of our work. Finally, we design a tailored bidirectional recurrent reconstruction framework specifically for spike cameras.
>
> **Q6:  Why not leverage any of the wavelet structure from WGSE?**
>
> The wavelet structure in WGSE (Zhang et al., 2023). exists solely within its representation. We replaced our representation with that of WGSE and retrained it. As shown in Table.3, the performance of our method (PSNR: 45.075 and SSIM: 0.9868) is better than that (PSNR: 42.302 and SSIM: 0.9744).

---

> ### Author Response · Authors · 2023-11-21
>
> Thank you for reading! We responded later than expected due to the substantial content additions. However, with the deadline approaching, we sincerely hope all reviewers to respond, allowing us to continue providing further information.

---

> ### Author Response · Authors · 2023-11-23
>
> We appreciate the professionalism and fairness of the official reviewers. In the process of responding to your questions, we have also gained insights on how to improve our paper. We sincerely hope that all the reviewers will make a fair and unbiased decision regarding the outcome of this article. If any of the public reviewers can disregard the results presented in the paper, the images and videos in the supplementary material, or even the promised release of model code and datasets, the quality of the open review environment would be greatly diminished.

---

### Official Review · Reviewer_rqug · 2023-11-05

**Soundness:** 2 fair
**Presentation:** 1 poor
**Contribution:** 3 good
**Rating:** 5
**Confidence:** 4

**Summary:**

This paper proposed a novel frame reconstruction method for spike camera. It particularly emphasized its advantage for light robustness and has shown results compared to previous methods within the same category. There are two areas of contributions claimed. The first is that the paper proposed a benchmark for high speed low light scenes. The dataset LLR is built upon existing method SPCS [Hu et al. 2022] and dimmed the scene brightness to obtain a low light version. The second contribution is an algorithm, including a light-robust representation, that leverages neighbor binned spike features to perform frame reconstruction.

**Strengths:**

This paper covers both dataset simulation and architectural proposals for low light. The proposed architecture is a push-forward based on previous transforms. I think the key idea is to extend existing LISI (local inter-spike interval) transform to incorporate the release time of forward and backward spikes. The reasoning is that as light intensity decreases, the spike interval increases, and it is well likely that information is helpful from longer time steps and bidirectional.

**Weaknesses:**

There are key issues associated with the proposal.
- First, the paper has not established benchmarks for the light robustness. It is very unclear how low is the "low light" used in this paper. And it's also not touched how robust the algorithm functions comparing normal and low light. A better version is a quantification for performance vs light intensity.

- Second, as light decreases, the solution of this paper is to extract information from longer time range. In such a case, motion may play a significant role affecting the reconstruction results. Yet it was not demonstrated.

It looks like the LLR dataset has only two lighting conditions, i.e. normal and low? Is it enough for benchmarking? Are 5 motions enough? The paper mentioned "... the power of light source is consistent with the real world". How to achieve consistency? The dataset part lacked technical details and justification.

The overall idea is interesting but lacks significance. The bidirectional attentive approach has been well seen in video frame interpolation and event-based version.
The global inter-spike interval (GISI) is a small extension of previous LISI. It is also very confusing what LISI is referring to. The two references [Chen 2022] proposed TFI and [Zhao 2022b] proposed DSFT (differential of spike firing). Are the authors referring to TFI and DSFT as the same thing? And according to Table 2 the significance of GISI is so marginal and is hardly considered a contribution.

The figures are well-made but they hardly explained technical details. Figure 5 generated a lot of confusion as what's "LISI", "update" and "maintain". Mathematical formulation is needed. Figure 7 did not provide useful information and is quite redundant to present after Eq 5-9.

Please work on the presentation as there are a lot of grammar errors.

**Questions:**

From Figure 11, it seems that the method converges well at 5 frames and even has worse results comparing 21 to 13 for PSNR. Is this contradicting to the choices for frame numbers?

I couldn't find where exactly is noise being handled. I only see an I_{dark} on top of I as in current but is that all?

---

> ### Author Response · Authors · 2023-11-20
>
> Thanks for your careful reading and valuable feedback!
>
>
> **Q1: Setting of light source power and How to achieve consistency with the real world?**
>
> We set the lighting parameters in the advanced 3D graphics software, blender, to make the lighting conditions as consistent as possible with the real world. The following are the configuration details in Blender.
>
> In Blender, various types of lighting simulation functions, including sunlight, point lights, and area lights, have been integrated into the graphical interface. We can adjust lighting parameters to control brightness and darkness.
>
> 1. For sunlight in Blender, the watts per square meter can be modified. Typically, 100 watts per square meter corresponds to a cloudy lighting environment. For the 'Car$_L$' scene in Fig.13, we have set sunlight to 10 watts per square meter, which is deemed sufficiently low.
>
> 2. For point lights and area lights, Blender allows modification of radiant Power, measured in watts.  This is not the electrical power of consumer light bulbs. A light tube with the electrical power of 30W approximately corresponds to a radiant Power of 1W. In the 'Cook$_L$' scene  in Fig.13, we have set an area light with the radiant Power to 1W (the electrical power of 30W) . It already represents a very dim indoor light source.
>
> **Q2:How dark is our dataset?**
>
> The low light scene in our dataset cover various dark range. The brightness is not only determined by the light source, but also by factors such as camera distance, object occlusion, and so on. These factors are ultimately reflected in the grayscale of the rendered image. Therefore, we calculate the grayscale histograms of images in low light scenes. As shown in **Fig.14** (each bar represents 5 grayscale levels). We can see that the grayscale is diverse and in a lower range.
>
> **Q3: A quantification for performance vs light intensity.**
>
> We generated spike streams under different lighting conditions by modifying the light source parameters. Due to the large time-consuming of once rendering for each type of lighting, we only use one scene to demonstrate the performance of our method. Related results are updated in **Table.4**. Our method demonstrates excellent performance.
>
>
> Specific modification details: We set sunlight (see Q1) to 30, 50, 70, 90, 110 and 130 watts per square meter to render images respectively.
>
>
> **Q4: Diversity of motion**
>
>
> The motion in LLR are diverse. Each designed scene contains 1000 frames of images and the motion at different times is different. We generate a optical flow every 40 frames for LLR. The degree distribution of the optical flow is statistically analyzed in **Fig.15**. We can find that the motion in LLR covers all kinds of directions.
>
>
> **Q5: The significance of our overall idea**
>
> We appreciate your recognition that our idea is interesting. The bidirectional RNN itself is a meaningful structure and emerges in the NLP domain. Many other domains, such as video interpolation, multi-agent systems, and event-based vision, have been inspired and modify the structure to better suit their field. We follow the same research line, and we are the first ones that design a tailored bidirectional recurrent reconstruction framework specifically for spike cameras. Our experimental results demonstrate that RNN structure can also boost the performance in spike vision.
>
>
> **Q6: TFI \& DSFT**
>
> Both TFI and DSFT first calculate the interval between adjacent spikes in the input spike stream, and they both employ a brute-force search approach. The difference lies in the fact that TFI takes the reciprocal of each interval.
>
>
>
>
>
>
> **Q7: LISI VS. GISI**
>
>
> GISI (our final method) not only outperform LISI (baseline (E) in Table.2) in both PSNR and SSIM on synthetic datasets but also have better generalization  on real spike streams.
>
>
> We update reconstruction results of two methods in **Fig.17**. More importantly, the cost of using GISI instead of LISI is negligible (we only need to use two 400x250 matrices to store the time of the forward spike and the backward spike, respectively), which does not affect the parameter and efficiency of the network at all.
>
>
>
> **Q8: Fig.5 \& Fig.7**
>
>
> Thanks for your suggestions.
>
> 1. For **Fig.5**, we have included additional supplementary explanations for LISI in the appendix and highlighted them in the main paper. The "maintain" and "update" are unique operations in GISI. For better understanding, it is suitable to be written as pseudocode. The pseudocode has been written in the appendix. In the caption of Fig.5, we have updated the link for the specific location of the algorithm.
>
> 2. For **Fig.7**, we have also adjusted its positioning to better illustrate the specific modules of the network, placing it before the formulas.

---

> > ### Author Response · Authors · 2023-11-20
> >
> > **Q9: Grammar errors**
> >
> > We have corrected the relevant Grammar errors.
> >
> > **Q11: From Fig.11, it seems that the method converges well at 5 frames and even has worse results comparing 21 to 13 for PSNR. Is this contradicting the choices for frame numbers?**
> >
> > Our method is robust to the setting of the number of spike streams within a certain range, i.e., when the number of spike streams exceeds a certain value, the method performance can converge (with certain fluctuations). Therefore, as long as the selected hyperparameter is within the appropriate range for test, it is reasonable.
> >
> >
> > **Q12: Noise of spike cameras**
> >
> > We use the spike camera simulator  (Zhao et al. 2022a) to generate data. It takes into account Poisson noise and fixed-pattern noise. By doing so, we believe our data is closer to that from the real scenes since noises are inevitable in the real world. In addition, training on data with noises can help the model avoid overfitting as shown in **Fig.16**. Besides, we never claim our method aims to deal with the noise problem as a contribution, this discussion is out of the scope of the main paper.

---

> ### Author Response · Authors · 2023-11-21
>
> Thank you for reading! We responded later than expected due to the substantial content additions. However, with the deadline approaching, we sincerely hope all reviewers to respond, allowing us to continue providing further information.

---

> ### Author Response · Authors · 2023-11-23
>
> Thanks for taking the time to read. We have provided lots of additional results to address your concerns. Additionally, we have provided explanations regarding the questions raised by the public reviewers. We would greatly appreciate your feedback.

---

> ### Author Response · Authors · 2023-11-23
>
> We appreciate the professionalism and fairness of the official reviewers. In the process of responding to your questions, we have also gained insights on how to improve our paper. We sincerely hope that all the reviewers will make a fair and unbiased decision regarding the outcome of this article. If any of the public reviewers can disregard the results presented in the paper, the images and videos in the supplementary material, or even the promised release of model code and datasets, the quality of the open review environment would be greatly diminished.

---

### Public Comment · ~S.Ahmad1 · 2023-11-21
**An Interesting Work, but with Several Serious Issues (2)**

1. **Incomplete Comparison:** The authors claim that “Existing methods struggle to perform well in low-light environments due to insufficient information in spike streams.” However, as far as I am aware, **references such as Ref1, Ref2, and Ref3 have researched the issue of spike-to-image reconstruction under low light conditions**. (Note that Ref1 also proposed a Similar Recurrent Structure and has open-sourced their codes). **Why were these low-light enhanced spiking reconstruction methods not compared (or even cited) in this paper?** Instead, the chosen methods for comparison do not specifically target low-light enhancement. Could this demonstrate the superiority of the proposed method?

    *Ref1: Recurrent Spike-based Image Restoration under General Illumination, 2023*

    *Ref2: High-Speed Scene Reconstruction from Low-Light Spike Streams, 2022*

    *Ref3: Reconstructing Clear Image for High-Speed Motion Scene with a Retina-Inspired Spike Camera, 2021*

2. **Concerns about Experimental Results:** The proposed method and the ones  (e.g., SSML, S2I, WGSE) compared in this paper achieve a PSNR (Peak Signal-to-Noise Ratio) of over 35, as also mentioned by Reviewer ajbr. **It is commonly known that when PSNR exceeds 35, human visual sensitivity can hardly distinguish differences in image quality. However, the experimental results illustrated in Fig. 10, where authors had 30 individuals score the visual quality of the reconstructed images, show a significant difference among the proposed method and others** This outcome seems quite unbelievable!! Could you please explain the reason?

3. **Concerns about Experimental Results:** On the same datasets (e.g., the dataset of Zhao et al., 2021; the dataset of Dong et al., 2022), **the visual quality of the experimental results in this paper appears much worse (more noise, less clarity in details) compared to those results in their original papers** (e.g., S2I, SSML, STP, SNM, WGSE). This is very confusing! Could you please explain the reason for this? Is it possible to open source the experimental settings and checkpoints for reproducing the results of these comparative experiments? I somewhat suspect that you **did not well-train** other methods on these datasets (ˆ_ˆ).

4. **Concerns about Experimental Results:** Experimental results of proposed method, as shown **in Tables 1, 2, 3, 4, and 5, where PSNR values consistently surpass 40 (even exceeding 45), are incredible compared to existing low-light spike-to-image works (Ref1, Ref2, Ref3)!** Particularly, the visualized results of proposed method seem to show no substantial improvement over existing similar works. Could you please release codes and the model checkpoints for reproducing these results? Considering the convenience of Open Review System, we could hardly trust the validity of these experimental results, because we have no access to the datasets and model ckpts.

5. **Concerns about the Speed and Domain Gap of Dataset:** The main contribution of this paper is the synthetic spike dataset for high-speed low-light scenes. However, **I did not find any details introducing the “speed” of moving objects in each scene of the simulated dataset**. Moreover, there appear to be no experiments conducted to measure the difference between synthetic data and real spike data.

6. **Unconvincing Motivation:** As the author mentioned in the paper: “with the decrease of illuminance, the total number of spikes in spike streams decreases greatly, indicating that the valid information also decreases significantly”. **This suggests that spike cameras may be not suitable for low-light high-speed scenarios**. Therefore, why choose spike cameras rather than event cameras (with a higher dynamic range) or high-speed cameras (e.g., i-SPEED 7, Phantom T4040) for this task? **Are there any experiments in this paper supporting the research motivation or necessity**?

7. **Limited Contribution:** (1) The claimed core contribution (i.e., the proposed GISI, a key component of LR-Rep) is merely to calculate the inter-spike-interval (Zhao et al. 2020b) in the spike stream between [t_i, t_i+1]. **This appears to be a trivial operation, as also raised as a concern by Reviewer rqug**. (2) Furthermore, the proposed fusion module merely utilizes the Pyramid Cascaded Deformable convolution (PCD) (Wang et al., 2019) to fuse features extracted from spike streams of [t_i, t_i-1] and [t_i, t_i+1]. This seems hardly novel, **particularly since the cascaded deformable structure has been extensively applied in spike-to-image reconstruction (Ref4, Ref5)**. (3) The synthetic dataset LLR contains only 5 scenes, which seem quite simplistic and toy-like.

    *Ref4: Learning to Reconstruct Dynamic Scene from Continuous Spike Stream, 2021*

    *Ref5: Learning to Super-Resolve Dynamic Scenes for Neuromorphic Spike Camera, 2023*

---

> ### Author Response · Authors · 2023-11-22
>
> **1. Incomplete Comparison:**
>
> We have compared our method with a sufficient number of methods to demonstrate its effectiveness.
>
> Ref1 is from the MM conference, which has the same timeline as ICLR. This is a concurrent piece of work.
>
> Regarding ref2, we have actually tested the method, but it struggles to handle scenes with a large motion. We have included it in the supplementary material.
>
> As for ref3, it did not specifically design modules to handle low-light data or test low-light scenarios. Besides, ref3 is not the state-of-the-art method even for the normal-light data. No work includes the comparison with all methods in their field.
>
> **2. Concerns about Experimental Results**
>
> We have addressed and updated the information about PSNR in our response to Reviewer ajbr's comments. Please refer to that for clarification.
> In our user study, we adjusted the brightness uniformly first for viewing (details can be found in our paper by searching for the keyword "apply"). Then, they are presented to 30 individuals independently. We assure you of their authenticity.
>
> **3. Concerns about Experimental Results**
>
>
> We have provided the relevant information on training S2I and WGSE in the main paper. Here, we further clarify more details.
>
> For TFSTP, SNM, and SSML, we use their open-source codes for testing!  The dataset of Zhao et al., 2021 is also open-source. You are welcome to reproduce the results to verify the authenticity of the results instead of questioning us.
>
>
> For all the supervised deep learning methods, i.e., S2I and WGSE, they were trained on the RLLR, consistent with our method to ensure fairness. S2I and WGSE offer open-source source codes. We trained and tested based on the source codes and the relevant parameter settings are the same as their paper!
>
> Potential sources of visual differences are as follows:
>
> 1. Different ways of generating spike streams between RLLR and the training set from S2I: The training set in S2I uses an ideal simulation without noise to generate spike streams. However, in RLLR, we used the state-of-the-art spike camera simulator (R1) to generate spike streams, which introduces diverse noise including temporal noise and fixed-pattern noise. This introduces more difficulties in training for other methods to converge. However, our bidirectional recurrent-based reconstruction method can effectively aggregate more temporal information. Since our structure is suitable for addressing the noise in the dataset, our method can converge better. In addition, we also discussed the impact of the presence or absence of noise in training set on the results in our appendix (Fig.16).
>
>
> 2. Different frames used for comparison: The displayed results in our paper are not at the same time as those in the cited work. Many methods show temporal fluctuations in their reconstructions, while ours is more stable. You can refer to the supplementary materials for the reconstruction video.
>
>
> 3. Differences in position and scale: For reconstructed results of different methods, we have selected a more differentiated and representative location to enlarge and show.
>
> We guarantee the reproducibility and authenticity of our experiments under the supervision of all public reviewers, official reviewers, AC, and PC. Besides, after publication, we will release all datasets and the checkpoints for S2I and WGSE.
>
> **4.  Concerns about Experimental Results**
>
> This question overlaps with the previous ones. Please refer to Q2 for clarification on the higher PSNR of our method. Regarding reproducibility, as stated at the beginning of the paper, we will provide all codes and datasets after publication.
>
> **5. Concerns about the Speed and Domain Gap of Dataset:**
>
> The speeds in the scenes are set to mimic real-world scenarios. Taking the car scene as an example, the speed set in Blender is approximately 70 km/h. To provide more insights, we calculated the distance between two consecutive positions based on the designed action, which is approximately 5.099e-4m. Further, 5.099e-4 * 40000 * 3600 = 73 km/h. We have included an image in the supplementary material to illustrate these details.

---

> ### Author Response · Authors · 2023-11-22
>
> **6. Motivation \& "spike cameras may be not suitable for low-light high-speed scenarios":**
>
>
> We cannot understand why this conclusion is inferred from our paper. Various sensors experience a decrease in data quality under low-light conditions, including DVS (R2). Following your logic, should we also conclude that R2 indicates DVS is not suitable for low-light scenes since it designed the architecture for low-light conditions?
>
> Spike camera has shown enormous potential for high-speed visual tasks, such as reconstruction, optical flow estimation, and depth estimation (A large amount of work is accepted by high-quality meetings and you can find citations in our main paper). However, we have observed a challenge, i.e., the spike camera faces the decreasing data quality as other sensors under low-light conditions. Hence, this paper aims to address the challenges of the spike camera and our method successfully accomplishes them.
>
> **7. Limited Contribution:**
>
> In addition to the proposed contribution, you overlooked our other contribution. All the reviewers rated our contributions as 'good.' According to your argument, most papers would not be novel, as they can trace influences from others. The novelty lies in whether new problems are addressed (low-light reconstruction in spike vision), whether a reasonable framework is designed for new problems, and whether results are achieved that other methods cannot reach. If addressing an unattended problem is not considered novel, then what really is novelty?
>
>
> R1. Junwei Zhao, Shiliang Zhang, Lei Ma, Zhaofei Yu, and Tiejun Huang. Spikingsim: A bio-inspired
> spiking simulator. In 2022 IEEE International Symposium on Circuits and Systems (ISCAS), pp.
> 3003–3007. IEEE, 2022a.
>
> R2. Rui Graca, Brian McReynolds, and Tobi Delbruck. Optimal biasing and physical limits of dvs event
> noise. arXiv preprint arXiv:2304.04019, 2023.

---

> > ### Public Comment · ~S.Ahmad1 · 2023-11-23
> > **Thanks for the author’s response (updated)**
> >
> > + We sincerely appreciate the authors' response! However, these replies have not addressed our concerns. **We wish to re-emphasize that this is not an academic attack!** There is no necessity to target a paper with limited impact. It is important to note that the Open Review platform was founded with the goal of promoting openness in scientific communication, especially in the peer review process. **Isn't enduring the rigour of peer review a fundamental criterion for a "qualified" academic work?**
> >
> > **1**.Why were these low-light enhanced spiking reconstruction methods not compared (or even cited) in this paper? **This issue remains unresolved.**
> >
> >    The following three methods all handle the low-light spike-to-image challenge:
> >
> >    (1)**Ref1** was open-sourced on August 6, 2023, nearly two months prior to the ICLR 2024 submission deadline..
> >
> >    (2)**Ref2:** the low-light enhancement method for spike camera was not evaluated in the experiments of this main paper.
> >
> >    (3)**Ref3** states that "When the light becomes weak, it produces a sparse spike stream with longer inter-spike intervals. ...  due to the existence of object occlusion or illumination changes, the assumption of temporal consistency of light intensity along motion trajectories can be non-reliable." To address this, Ref3 proposed the Motion-Aligned Temporal Filtering model.
> >
> > We agree with the authors that " No work includes the comparison with all methods in their field." However, the pivotal question remains: why does the comparison not involve the state-of-the-art **low-light** spike-to-image reconstruction method, but rather other methods that do not specifically target this task?
> >
> > **2**.It is commonly known that when PSNR exceeds 35, human visual sensitivity can hardly distinguish differences in image quality.
> >
> > However, the experimental results shown in Fig. 10, where 30 individuals scored the visual quality of the reconstructed images, present a significant difference between the proposed method and others. **Why are the experimental results so significant when image quality is approaching or even exceeding human visual sensitivity?**
> >
> > **3**.As the authors stated the “visual differences” in experimental results between this paper and the original papers of other methods:
> >
> > (1)Taking Figure 9 in the paper as an example, these results are conducted on a public **real** spike dataset, which does not require the use of the **simulator** (Zhao et al. 2022a) for data generation. **Hence, there is no data discrepancy caused by the simulator.**
> >
> > (2)The authors clarify that “Many methods show temporal fluctuations in their reconstructions,” which may be a reason the visual results in this paper are worse than in their original papers. **However, this issue is not mentioned in the official papers of S2I, SSML, STP, SNM, WGSE.**
> >
> > **4**.Can the model checkpoints (ckpts) for reproducing the experimental results in the paper be released?
> >
> > **5**.The proposed LLR dataset contains 5 scenes, namely Ball, Car, Cook, Fan, and Rotate. **We could not find any cars in the car scene, so how is the moving object determined to be a car?** Besides, the authors claim that this is a **high-speed** dataset mimicking real-world scenarios. **However, in real-world scenarios, how "high-speed" can activities like cooking in a pot or playing pool on a table be?**
> >
> > Moreover, there appear to be no experiments conducted to measure the difference between synthetic data and real spike data. **The similarity degree between the proposed synthetic dataset and real spike data cannot be determined.**
> >
> > **6**.As stated in the paper: "With the decrease of illuminance, the total number of spikes in spike streams decreases greatly, indicating a significant reduction in valid information." This observation highlights the degraded performance of spike cameras under low-light conditions. **In contrast, while event cameras become noisier in low-light conditions, they still maintain a competitive edge over most visual cameras, thanks to their high dynamic range (>140 dB) [Ref6]. Why weren't comparative experiments conducted with event cameras in this paper?** Are there any experiments in this paper supporting the research motivation or necessity?
> >
> > *Ref6: Learning to See in the Dark with Events. ECCV. 2020*

---

> > > ### Author Response · Authors · 2023-11-23
> > >
> > > Please don't hide behind these high-sounding reasons to disguise the fact. Whether you are being stringent or nitpicking can be judged from your words. You have deleted and resent the previously posted questions. In fact, in the our new response, we have already provided further explanations.
> > >
> > > **5. LLR**
> > >
> > > The car describes shooting outside the window while driving. Regarding the design of motion, we are based on Blender. One of the most commonly used operations is to define key frames. We simulate the motion of the car by designing the position and angle of objects at key frames. Due to time constraints, you can refer to the Blender official documentation to learn.
> > >
> > > **6. DVS**
> > >
> > > Please note that this paper focuses on solving the challenge encountered by spike cameras rather than event cameras.

---

> > > > ### Public Comment · ~S.Ahmad1 · 2023-11-23
> > > > **New Public Comments**
> > > >
> > > > **7.Limited Novelty**. Apart from the contributions outlined in the paper, specifically, (1) a reconstruction dataset for high-speed low-light scenes, and (2) a recurrent-based reconstruction framework (including a light-robust representation LR-Rep, and a fusion module), **we are unclear about the "other contributions" mentioned in authors’ recent response**?
> > > >
> > > > Regarding the main contributions:
> > > >
> > > > (1) The synthetic dataset LLR comprises merely 5 scenes, which appear overly **simplistic and toy-like**. Moreover, dataset details regarding the speed and domain gap between real spike data and synthetic data are not provided, potentially limiting its impact on the community.
> > > >
> > > > (2) The proposed GISI, a key component of the LR-Rep framework, merely computes the inter-spike interval (Zhao et al. 2020b) within the spike stream spanning the interval [t_i, t_{i+1}]. **This procedure is elementary and scarcely represents a contribution**. Furthermore, the approach of utilizing spike information before and after the precise moment (t_i) to enhance image reconstruction quality is widely used in the spike vision community [Ref7, Ref8, Ref9], and thus, **does not present as a novel contribution**. (Additionally, given that spike intervals vary with different illumination conditions, why is the temporal window in the GISI transform fixed?)
> > > >
> > > > *Ref7: learning to Reconstruct Dynamic Scene from Continuous Spike Stream. 2021*
> > > >
> > > > *Ref8: Learning to Super-resolve Dynamic Scenes for Neuromorphic Spike Camera. 2023*
> > > >
> > > > *Ref9: Learning Optical Flow from Continuous Spike Streams. 2022*
> > > >
> > > > (3) The fusion module employs Pyramid Cascaded Deformable convolution (PCD) (Wang et al., 2019) to merge features extracted from spike streams of [t_i, t_i-1] and [t_i, t_i+1], **which hardly seems novel**, particularly given the extensive application of the cascaded deformable structure in spike-to-image reconstruction (Ref4, Ref5).
> > > >
> > > > Concerning the **reasonableness** of the proposed framework, **we share similar concerns with Reviewer rqug** that "motion may significantly affect the reconstruction results when the input spike stream covers a longer time range." **However, the authors have not addressed this.**
> > > >
> > > > Regarding the authors' claim that "If addressing an unattended problem is not considered novel, then what really is novelty?" We believe the authors could **overclaim** their contribution, as several existing efforts have studied the low-light spike-to-image reconstruction (Ref1, Ref2, Ref3). Furthermore, similar Recurrent Structures and Pyramid Cascaded Deformable convolutions **have been extensively utilized in prior research**.

---

> > > > > ### Author Response · Authors · 2023-11-23
> > > > >
> > > > > **7.1 The synthetic dataset LLR**
> > > > >
> > > > > LLR includes 5 types of motion and 2 types of light sources. Note that  these are carefully designed. We can certainly generate a large number of random scenes like RLLR. But that's not suitable for evaluation.
> > > > >
> > > > > **7.2 GISI**
> > > > >
> > > > > GISI (our final method) not only outperform LISI (baseline (E) in Table.2) in both PSNR and SSIM on synthetic datasets but also have better generalization on real spike streams.
> > > > >
> > > > > We update reconstruction results of two methods in Fig.17. More importantly, the cost of using GISI instead of LISI is negligible (we only need to use two 400x250 matrices to store the time of the forward spike and the backward spike, respectively), which does not affect the parameter and efficiency of the network at all.
> > > > >
> > > > > **7.3 The fusion module**
> > > > >
> > > > > The proposed structure is different from the cited article. We combine a residual operation to provide alignment information for bidirectional temporal features and then uses ResNet to extract deep features.

---

> > > > > > ### Author Response · Authors · 2023-11-23
> > > > > >
> > > > > > Due to time constraints, we may not be able to provide you with more detailed information. But I hope everyone can maintain a fair attitude and think whether most of the confusion has been resolved.

---

> ### Author Response · Authors · 2023-11-23
>
> Despite the limited time available before the deadline, we have efficiently and to the best of our ability responded to your concerns.   I am starting to suspect that this is an academic attack rather than a normal exchange.
>
> First, we must express our dissatisfaction with your comments, which we find unprofessional and biased. Firstly, as you acknowledged, you are not a formal reviewer nor a representative figure in the field, which implies that your concerns might be based on misunderstandings. Therefore, your expression should be 'doubts' rather than 'serious issues,' which leans towards a negative connotation that could influence the reviewers and readers. Secondly, given that you claim to be a researcher in neuromorphic vision, your declaration of no conflicts of interest is meaningless.
>
> Second, We are happy to answer your questions, but before we respond, the words you use, like 'hardly trust the validity of these experimental results,' 'somewhat suspect that you did not well-train other methods on these datasets,' and 'This outcome seems quite unbelievable!!' carry a very strong personal emotion. With such strong subjective beliefs, we are puzzled whether you would be willing to be convinced, even if we presented solid evidence.
>
> Third, do you not understand the meaning of 'concurrent work'? Why isn't work from two months ago considered concurrent? Also, whether to compare concurrent works should be judged by the reviewers. You can ask questions, and we have given you the reasons. We think continuing nitpicking ignoring the previous explanation is disrespectful and carries clear offensive intent. As for R3, this work does not provide experiments to support it can work under a low-light environment itself, however, you ask us to conduct the experiments to verify this? As for R2, the main content is limited, if this work is not competitive in the settings, why should we consider it on the main content?  But we can pose it.
>
>
> We hope to have a polite discussion based on basic facts, not personal emotions, but scientific rationality. We believe it would contribute positively to an open review environment.
>
> Please wait...

---

> ### Author Response · Authors · 2023-11-23
>
> **1. Relevant instructions for Ref1, Ref2, and Ref3.**
>
> Please refer to the fourth paragraph of the previous comment for details. **In addition, the results of Ref2 have been added to the supplementary materials.** We hope you can observe their quality with an objective attitude.
>
> **2. Relevant instructions of PSNR.**
>
> We actually answered this question in our previous response. We are willing to provide a more detailed explanation once again.
>
> (1) High PSNR:
>
> When the scene is so dark, both the Ground Truth (GT) and the reconstructed images can be at a lower grayscale level. This results in a significantly lower MSE compared to normal-light scenes. Consequently, their PSNR is large.
>
>
> The Peak Signal-to-Noise Ratio (PSNR) can be expressed as $PSNR = 10 \cdot \log_{10}\left(\frac{{\text{255}^2}}{{\text{MSE}}}\right),$
> where MSE represents the Mean Squared Error. It can be further defined as,
> $MSE = \frac{1}{mn} \sum_{i=0}^{m-1} \sum_{j=0}^{n-1} \left( I(i, j) - K(i, j) \right)^2,$where m (n) is the height (width) of the image, respectively and $I(i, j)$($K(i, j)$) represents the pixel value at (x, y) in ground truth (the reconstructed image). Hence, when both the Ground Truth (GT) and the reconstructed images are at a lower grayscale level, PSNR can be large. In Fig.8, the scene is exceptionally dark, and we applied the same gamma correction to the reconstructed result for better visibility to readers.
>
> (2) Calculate PSNR after scale the images to [0,255]:
>
> Based on the comments of Reviewer ajbr, we scaled the image to [0, 255] before calculating PSNR. The relevant results are as follows:
>
>
> ***
> | Metric         | TFI     | SSML    | S2I     | STP     | SNM     | WGSE    | Ours    |
> | -------------- | ------- | ------- | ------- | ------- | ------- | ------- | ------- |
> | PSNR           | 31.409  | 38.432  | 40.883  | 24.882  | 25.741  | 42.959  | **45.075** |
> | PSNR (scale)   | 21.665  | 30.176  | 31.202  | 14.894  | 18.527  | 32.439  | **38.131** |
>
> ***
>
>
> **You can see that our method has a PSNR (scale) of 38.131, while none of the other methods exceed 35! Therefore, the reconstruction results can be distinguished after scaling them to normal brightness!**
>
> (3) Quality of visualization results
>
> We have presented the reconstruction results of all methods in the main paper. Our reconstruction results are visually  better. This can be perceived by 30 invited individuals, us, and official reviewers. We hope that these results can be evaluated with an objective and fair attitude.
>
>
> Please wait...

---

> ### Author Response · Authors · 2023-11-23
>
> **3. experimental results**
>
> (1) dataset:
> Perhaps you have some misunderstandings here. In our first response (3.1), we mentioned that results from the same model trained on different datasets are different.
>
> (2) temporal fluctuations:
> We have provided a reconstruction video to showcase this situation in the supplementary materials, and we also informed you in our first response. Ignoring our answers and asking the same question is an unprofessional and irresponsible behavior.
>
> **4. The model checkpoints:**
>
> In our first response (3 and 4), we have informed you **more than once** that we will provide checkpoints, codes, and datasets after publication for reproducibility!
>
> Please wait...

---

### Public Comment · ~S.Ahmad1 · 2023-11-21
**An Interesting Work, but with Several Serious Issues (1)**

I am a researcher in the field of neuromorphic vision, and have been following the efforts of spike cameras for several years. I noticed this paper at this year's ICLR and found it to be quite interesting. This paper proposes a synthetic spiking dataset based on existing works (SPCS, Hu et al. 2022), and a spike-to-image reconstruction method. However, after a careful reading, several serious issues left me concerned.

+ **Please note that my concerns have little influence on the acceptance of this paper because I am not an official reviewer, and I have no conflicts or intentions of aggression**. However, I still hope the authors could clarify my concerns. The presence of these serious issues really leads me to be concerned. I look forward to the authors' response. Thank you all very much.

---

### Public Comment · ~S.Ahmad1 · 2023-11-23
**Factual Errors existing in author provided results**

In the spirit of academic rigour, innovation, and reliability, we believe it necessary to raise these concerns, albeit with some reservations. We have observed potential **factual errors** in the experimental results presented in Table 1, specifically in the author's response to Reviewer ajbr ([https://openreview.net/forum?id=c0kTH3HVLz&noteId=KpW795r254](https://openreview.net/forum?id=c0kTH3HVLz&noteId=KpW795r254)). Our analysis is detailed below:

Given that $PSNR = 10 \log\left(\frac{255^2}{MSE}\right)$, and $MSE = \frac{1}{mn} \sum_m \sum_n (I_{m,n} - K_{m,n})^2$.

Suppose that the max pixel value in the ground truth image (under low-light conditions) is $K_{max}$ (where $K_{max} < 255$), and both the reconstructed image and ground truth image are scaled to [0,255], then the scaled $MSE$ becomes $MSE_s = \frac{1}{mn} \sum_m \sum_n \left(\frac{(I_{m,n} - K_{m,n})}{K_{max}} \times 255\right)^2$. Consequently, the scaled $PSNR$ becomes $PSNR_s = 10 \log\left(\frac{K_{max}^2}{MSE}\right)$. It leads to **$PSNR - PSNR_s = 10 \log\left(\frac{255^2}{K_{max}^2}\right)$, which is a constant**. This suggests that, in Table 1, the difference between each set of scaled results and original results should be a constant. However, the results shown in the table below provided by the authors are not consistent:

| Metric         | TFI    | SSML   | S2I    | STP    | SNM    | WGSE   | Ours   |
| -------------- | ------ | ------ | ------ | ------ | ------ | ------ | ------ |
| PSNR           | 31.409 | 38.432 | 40.883 | 24.882 | 25.741 | 42.959 | 45.075 |
| PSNR_s (Scaled) | 21.665 | 30.176 | 31.202 | 14.894 | 18.527 | 32.439 | 38.131 |
| PSNR - PSNR_s | 9.744  | 8.256  | 9.681  | 9.988  | 7.214  | 10.520 | **6.944** |

Such factual errors, along with the lack of datasets (including dataset details) and model codes (including experimental settings and checkpoints for reproducing results), **largely deepen our concerns about the unusual experimental results reported in the paper** (e.g., Concerns 2, 3, 4).

Finally, I greatly appreciate the authors' response. However, the controversies have not been addressed in a meaningful way. I would prefer to see these issues addressed directly in the research, rather than circumvented.

---

> ### Author Response · Authors · 2023-11-23
>
> Please note that the max pixel value varies in each scene, and the max pixel value also varies for each frame! You used a wrong assumption to come to a wrong conclusion.

---

> ### Public Comment · ~S.Ahmad1 · 2023-11-23
>
> The results presented in Table 1, visualized in Figure 8, are obtained from the Cook Scene. Under the specified scene (Cook) and the given frame (Figure 8), $K_{\text{max}}$ is fixed.

---

### Meta-Review · Area_Chair_zwyx · 2023-12-13

**Metareview:**

This paper proposes a frame reconstruction method for spike camera. It emphasized its advantage for light robustness and has shown results compared to previous methods within the same category. There are two areas of contributions claimed. The first is that the paper proposed a benchmark for high-speed low-light scenes. The dataset LLR is built upon the existing method and dimmed the scene brightness to obtain a low-light version. The second contribution is an algorithm, including a light-robust representation, that leverages neighbor-binned spike features to perform frame reconstruction.

(a) Strengths of the paper

- The novelty and relevance of the problem tackled, as low-light imaging with neuromorphic cameras is an emerging and significant challenge (Reviewer sPKH, ajbr, bgSj).

- The method's clear superiority over state-of-the-art methods in certain conditions and the inclusion of extensive ablation studies and validations (Reviewer sPKH).

- The comprehensive treatment of the dataset and the architecture designs for low-light environments (Reviewer bgSj).

(b) Weaknesses of the paper

There are concerns regarding the clarity and presentation of the paper, with multiple reviewers pointing out grammatical errors and unclear figures (Reviewers bgSj, rqg).

Questions about the robustness of the model under different lighting conditions and noise levels have been raised, suggesting that the forward model may not adequately represent real-world conditions (Reviewer sPKH).

The paper has been criticized for not providing access to reconstructed videos or sufficiently detailed demonstrations of the method's performance, which is crucial for evaluating the presence of artifacts such as flickering (Reviewer sPKH).

Some reviewers question the clarity of methodological descriptions, such as the conversion process from RGB to grayscale and the rationale behind high PSNR values (Reviewer ajbr).

Reviewers also noted a lack of detailed justification for the dataset and queried the representativeness of the lighting conditions used for benchmarking (Reviewer rqg).

The paper exhibits certain innovative aspects with its approach to spike camera image reconstruction. However, the reviews suggest that there are significant issues with the paper's presentation and clarity (bgSj, rqg) , coupled with concerns about the thoroughness of its experimental evaluation (sPKH, rqg) and methodological explanations (ajbr).

Taking all factors into account, the area chair does not recommend this paper be accepted at ICLR because of experimental valiadation and methodological explanations.

**Justification For Why Not Higher Score:**

The paper exhibits certain innovative aspects with its approach to spike camera image reconstruction. However, the reviews suggest that there are significant issues with the paper's presentation and clarity, coupled with concerns about the thoroughness of its experimental evaluation and methodological explanations.

**Justification For Why Not Lower Score:**

N/A

---

### Decision · Program_Chairs · 2024-01-16

Reject